# Metal Oxide Nanoparticles: Review of Synthesis, Characterization and Biological Effects

**DOI:** 10.3390/jfb13040274

**Published:** 2022-12-05

**Authors:** Andreea Mariana Negrescu, Manuela S. Killian, Swathi N. V. Raghu, Patrik Schmuki, Anca Mazare, Anisoara Cimpean

**Affiliations:** 1Department of Biochemistry and Molecular Biology, Faculty of Biology, University of Bucharest, 91-95 Splaiul Independentei, 050095 Bucharest, Romania; 2Department of Chemistry and Biology, Chemistry and Structure of Novel Materials, University of Siegen, Paul-Bonatz-Str. 9-11, 57076 Siegen, Germany; 3Department of Materials Science WW4-LKO, Friedrich-Alexander University, 91058 Erlangen, Germany; 4Regional Centre of Advanced Technologies and Materials, Palacky University, Listopadu 50A, 772 07 Olomouc, Czech Republic; 5Chemistry Department, King Abdulaziz University, Jeddah 80203, Saudi Arabia; 6Advanced Institute for Materials Research (AIMR), National University Corporation Tohoku University (TU), Sendai 980-8577, Japan

**Keywords:** metal oxide nanoparticles, nanotechnology, pro-regenerative potential, cancer therapy, antimicrobial activity, nanotoxicity

## Abstract

In the last few years, the progress made in the field of nanotechnology has allowed researchers to develop and synthesize nanosized materials with unique physicochemical characteristics, suitable for various biomedical applications. Amongst these nanomaterials, metal oxide nanoparticles (MONPs) have gained increasing interest due to their excellent properties, which to a great extent differ from their bulk counterpart. However, despite such positive advantages, a substantial body of literature reports on their cytotoxic effects, which are directly correlated to the nanoparticles’ physicochemical properties, therefore, better control over the synthetic parameters will not only lead to favorable surface characteristics but may also increase biocompatibility and consequently lower cytotoxicity. Taking into consideration the enormous biomedical potential of MONPs, the present review will discuss the most recent developments in this field referring mainly to synthesis methods, physical and chemical characterization and biological effects, including the pro-regenerative and antitumor potentials as well as antibacterial activity. Moreover, the last section of the review will tackle the pressing issue of the toxic effects of MONPs on various tissues/organs and cell lines.

## 1. Introduction

In the last few decades, the field of nanotechnology has become one of the most active areas of customizable materials science [1], with wide practicability in various clinical applications, due mainly to the specific size-dependent properties exhibited by the resulting nanomaterials as a direct consequence of a controlled synthesis procedure [2]. Amongst the already in use nanomaterials, nanoparticles (NPs) have received a great deal of attention due to their small size and large surface area [3], properties which provide researchers with novel ways of diagnosing and treating diseases that prior to this were thought to be unapproachable due to the size limitations. With multiple advantages such as high stability, simple preparation methods, excellent engineering control over aspects ranging from size, shape, porosity, etc. and cellular penetration capability, MONPs have grown into valuable materials for the drug and health-related industry [4]. Through the design and development of engineered MONPs, the limitations imposed by their bulk counterparts could be finally overcome, allowing researchers to make astounding breakthroughs in fields such as specific drug delivery, bio-imaging, biomolecule sensors, etc. [5,6]. Moreover, due to their reduced size, metal oxide nanoparticles can interact on a more in-depth level with various cellular structures compared to their bulk counterparts, and, more importantly, they do not cause systemic toxicity due to their highly improved biocompatibility [5,6]. Currently, various types of MONPs are used in clinical practice as antibacterial and wound healing dressings, biosensors and anticancer and image contrast agents [7]. Of these, zinc oxide NPs (ZnO NPs), cerium oxide NPs (CeO_2_ NPs), iron oxide NPs (Fe_2_O_3_ NPs), silver oxide NPs (AgO NPs), magnesium oxide NPs (MgO NPs), titanium oxide NPs (TiO_2_ NPs), nickel oxide NPs (NiO NPs), zirconium oxide NPs (ZrO NPs) and cadmium oxide NPs (CdO NPs) are the most promising candidates for biomedicine, with a considerable amount of research data available in recent literature regarding their biological in vitro and in vivo activity.

ZnO NPs are a nontoxic, biocompatible biomaterial, with unique abilities that may vary depending on their size, shape, orientation, morphology and aspect ratio [8]. They are widely used in commercial products such as sunscreens, ointments, food packaging and everyday-care products. Moreover, ZnO NPs exhibit a strong antibacterial effect, mainly attributed to their distinct characteristics, that is also dependent on dose, time and synthesis method [8] In addition, due to their inherent anticancer activity, ZnO NPs have been approved by Food and Drug Administration (FDA) as a new and potent antitumor therapy [9]. It is generally accepted that in addition to the generation of high levels of reactive oxygen species (ROS), ZnO NPs can exhibit a selective cytotoxic effect against cancer cells through the induction of an impaired equilibrium of zinc-dependent protein activity [10]. However, ZnO NPs were shown to induce toxic effects in different cells and organisms, thereby requiring further studies meriting their therapeutic benefits over the potential toxicological risk [11].

CeO_2_ NPs represent another type of MONPs widely used in biomedical applications due to their unusual antioxidant properties and anticancer activity. The basis for CeO_2_ NPs activities lies in the thermodynamic efficiency of redox-cycling between 3+ and 4+ states on their surface [12] and their unusual ability to absorb and release oxygen [13]. It is worth noting that these NPs can also exhibit pro-oxidant effects at lower pH values [14] and high concentrations [15], and data found in the literature suggest that, depending on their synthesis procedure, dosage and exposure time, they can induce cytotoxic effects [16].

TiO_2_ NPs are prevailingly used in bone and tissue engineering due to their ability to induce cell migration, adhesion, osseointegration and wound healing [17,18]. However, they also serve as an excellent antibacterial agent [19,20], playing an important role in bacteria growth inhibition via ROS production in the presence of ultraviolet (UV) light [21,22]. Moreover, due to their ability to generate high levels of ROS, they also exhibit anticancer activity [23].

One of the most promising nanoscale biomaterials is represented by Fe_2_O_3_ NPs, which can either be used as standalone agents (functionalized with other bioactive molecules/agents), embedded in composites, or bound to different types of cells [24]. Generally, Fe_2_O_3_ NPs are mostly used as drug-delivery platforms for pro-regenerative purposes or anticancer therapies, where the selectively targeted release of an active drug is accomplished by either the use of specific binding proteins or by the influence of external magnetic fields [25,26]. In addition, various cells can be magnetically marked with these NPs, therefore allowing for a non-invasive in vivo monitoring of the efficiency of a therapy [27,28].

MgO NPs are primary non-poisonous nanomaterials, with biomedical applicability as drug-delivery systems for anticancer therapy and antibacterial dressings, especially as, being soluble, adverse effects due to remaining in the tissue are avoided [29].

NiO NPs are a class of nanomaterials with a wide variety of flexible properties and vast applicability in the biomedical field through their antibacterial, antifungal and anticancer activities [30].

Because of their favorable biodegradable, mechanical and optical characteristics, ZrO NPs have piqued the interest of researchers, but despite their high biomedical potential, literature data regarding their role as antitumor, antibacterial and pro-regenerative materials are scarce [31].

Owing to their excellent antibacterial properties, which stem from their small size and morphology combined with the release of ions and ROS, CdO NPs play an important role in treating various bacterial and fungal diseases [32]. In addition, due to their unique physicochemical properties, CdO NPs exhibit better antitumor activity compared to any other heavy metal oxide nanoparticle, but only at lower concentrations [33].

Currently, MONPs are held at an enormous market value due to the significant improvements brought in various fields of nanobiomedicine and their immense potential for future applications.

The present review addresses in the first chapter, the various synthesis methods and characterization of the resulting MONPs, while the following chapters focus on the biological effects exhibited by the MONPs, particularly the pro-regenerative potential, anticancer activity and antibacterial properties. In addition to the positive aspects of MONPs, their very controversial biotoxicology is also discussed in detail alongside the future of these nanoparticles and if their positive therapeutic benefits can outweigh the potential risk caused by the toxic side effects.

## 2. Metal Oxide Nanoparticles: Synthesis, Characteristics, Surface Modification and Characterization

### 2.1. Synthesis Methods and Key Characteristics of Metal Oxide Nanoparticles

The synthesis of nanoparticles can be achieved with either a “top-down” or “bottom-up” approach. In the top-down approach, bulk materials are broken down into NPs by size reduction (via various lithographic techniques, milling, grinding, laser ablation, sputtering, etc.). In the bottom-up approach, NPs are obtained by chemical, physical and biological techniques (plant material, microbes, biological products, etc.).

Typically, chemical and physical synthesis routes (i.e., bottom-up approaches) are employed in the synthesis of MONPs, and result in an efficient quantity of obtained nanoparticles, but have the disadvantage of higher cost, presence of poisonous chemicals (e.g., absorbed on the NPs surface) leading to adverse effects when used in biomedical applications, the need to use of stabilizers, etc. [8,34,35,36].

Such synthesis routes include but are not limited to: (i) (chemical) precipitation, (ii) wet chemical synthesis, (iii) hydrothermal, (iv) solvothermal, (v) sol-gel, (vi) solid-state pyrolytic methods, (vii) thermal decomposition and (viii) microwave-assisted synthesis.

In order to overcome the disadvantages of MONPs synthesized through the usual classical routes that lead to adverse effects in biomedical applications, the green synthesis of MONPs, or biosynthesis, has gained significant attention due to the use of environmentally friendly and non-toxic reagents with diminished adverse/toxic effects, and result in increased biocompatibility. This approach includes, for example, the use of various biopolymers, plant leaf extracts, algae, surface active biosurfactants, etc. which offer higher specificity, biodegradability and biocompatibility [8,37,38]. Figure 1 shows an overview of the various synthesis routes used for manufacturing metal oxide nanoparticles.

Chemical precipitation is based, as the name suggests, on using a precipitating reagent (such as sodium hydroxide ammonium hydroxide or urea) in the metal precursor aqueous solution, and the resulting precipitate is annealed at high temperatures and converted to the corresponding MONPs. While such a synthesis method results in small-sized NPs, with a narrow distribution, and high purity, it can also lead to NPs with poor crystallinity and the risk of contamination can occur (from the intermediate formation). In the case of spontaneous precipitation, the process takes place without the addition of a precipitating reagent. Such precipitation methods are used for various MONPs, see for ZnO [39,40,41,42], CeO_2_ [43,44,45], Fe_2_O_3_ [46,47,48], TiO_2_ [49], MgO [50,51] and NiO [52]. The wet chemical synthesis method is based on the chemical precipitation method with the addition of an additive to stabilize the formed NPs; for example, such a synthesis method is used for ZnO NPs, with starch as the stabilizing agent [53], or for other MONPs (e.g., in the synthesis of CeO_2_ [54,55], Fe_2_O_3_ [56], TiO_2_ [49], MgO [57] NiO [58,59], ZrO [60] and CdO [61,62]).

Hydrothermal synthesis of MONPs is widely used due to the extensive control exercised over the morphology/particle size and lowering of particle aggregation, combined with suitability for large-scale production and high purity. Nevertheless, long reaction times are involved and several post-processing steps are required, as will be further discussed. The synthesis typically involves sealing a metal precursor aqueous solution into a Teflon line stainless-steel autoclave, together with a precipitating agent (e.g., NaOH) previously added dropwise to achieve the desired pH. The autoclave is then kept at a constant temperature (e.g., 80–200 °C) for a specific duration (e.g., 1–20 h), [63] followed by several washing steps and finally, annealing. Examples of hydrothermally grown MONPs include ZnO [64,65], CeO_2_ [66], Fe_2_O_3_ [67], TiO_2_ [68], MgO [69,70], NiO [71,72] and CdO [73].

The solvothermal synthesis is similar to the hydrothermal method, except other solvents are used in place of water. Typically, the reaction vessels or autoclaves are operated in a temperature range of 100 to 1000 °C and a pressure range of 1 to 10,000 bar [74]. The solvents used typically include diethanolamine (ZnO NPs [75]), ethanol (α-Fe_2_O_3_ NPs [76]), methanol (ZnO NPs [77]) 1,4-butanediol (γ-Fe_2_O_3_ NPs [78]), toluene (TiO_2_ NPs [79], NiO [80]) and ethylene glycol (Fe_3_O_4_ NPs [81]). Nevertheless, in some syntheses, the use of a stabilizer is necessary, and when targeting biomedical applications, it should also be biocompatible (e.g., trisodium citrate [81]).

The sol-gel method is a conventional and industrial method widely used for the synthesis of various NPs [82], offering, especially, good control over their size, high purity and homogeneity and low temperatures (on the downside, the use of organic solvents, availability of necessary precursors and long reaction times pose challenges). The key lies in the production of a homogeneous sol from the precursors and its conversion into a gel, followed by the removal of the solvent from the gel and subsequent drying. The molecular precursor is usually the corresponding metal alkoxide, which is dissolved in water or alcohol and converted to a gel by heating and stirring by hydrolysis/alcoholysis [82]. Appropriate drying methods are necessary depending on the desired properties and application of the resulting NPs. A noteworthy point is the broad size-distribution of particles obtained via sol-gel processes. Examples of MONPs synthesized by sol-gel include the synthesis of ZnO NPs either by a modified sol-gel method resulting in a 25 nm NPs, which is smaller than with previously reported sol-gel processes [41], or by typical sol-gel processes [83]. Additionally, several other MONPs can typically be obtained via the sol-gel method, e.g., α-Fe_2_O_3_ [84], MgO [63], NiO [85,86] and CdO [87].

Solid-state pyrolytic methods are based, as the name suggests, on the pyrolysis of the metal precursor, while the pyrolysis temperature controls the particle size and the additional chemicals and resulting by-products, and their dissolution can control the NP agglomeration [37]. For example, different sizes of ZnO NPs (8 to 35 nm) were obtained by adjusting the pyrolysis temperature of the reaction mixture [88].

MONPs obtained via thermal decomposition rely on heating the metal precursor above their decomposition temperature in a solvent with a high boiling point [89]. Such NPs have the advantage of being highly monocrystalline (i.e., a post-synthesis annealing is not necessary), but the yield of NPs is quite low. Typically, precursors are organometallic compounds dissolved in organic solvents, also containing surface-stabilizing agents, and the synthesis takes place at high temperatures in an inert atmosphere. Generally, an optimal precursor has to have a low decomposition temperature in order to result in a high surface area and low crystallite size. Moreover, high temperatures are avoided as they could lead to particle sintering, thus precluding the formation of NPs. The size of the NPs can be tuned through the reaction parameters, e.g., precursor, temperature, etc. This method is typically used in the synthesis of nanoparticles of ZnO [90], Fe_2_O_3_ [91,92], CeO_2_ [93], TiO_2_ [94], MgO [95], NiO [96,97] and CdO [98].

Microwave-assisted synthesis of nanomaterials has also gained tremendous ground, as the rapid heating of the reaction system can be achieved by microwave radiation, resulting in an enhancement of the reaction rate (a several orders of magnitude increase due to generation of localized reaction sites) and, thus, in a reduction in the reaction time [89]. Though such a synthesis method cannot be scaled up (no control over the temperature and pressure of the process), due to the high NPs formation yield, reduced agglomeration and fast reaction rates it is advantageous for further research. Various MONPs were synthesized using microwave-assisted techniques, see microwave polyol synthesis (ZnO [99], CeO_2_ [100]), microwave heating method (α-Fe_2_O_3_, β-Fe_2_O_3_, Fe_3_O_4_ [101], CdO [102]), solid-state microwave irradiation (NiO [103] NPs), microwave-assisted, solution-based synthesis of TiO_2_ [104,105] or NiO [106], microwave-assisted hydrothermal methods (ZnO [107]), low-power microwave-assisted heating (ZnO [108]), surfactant-free microwave-assisted mixing (ZnO [109]), etc.

In addition, there are other reported synthesis methods, generally not valid for the synthesis of all MONPs, but for specific metal oxides. These include mechanochemical synthesis (mechanochemical reactions in different milling times—Fe_2_O_3_ NPs [110]), co-precipitation via flow chemistry (Fe_2_O_3_ NPs [111]), continuous flow synthesis (TiO_2_ [112] or γ-Fe_2_O_3_ [113] NPs), successive ionic layer absorption and reaction (NiO [114]), direct chemical synthesis (NiO [115]), anodic arc plasma (NiO [116]) and so on.

The green synthesis of MONPs has received increased attention due to the use of environmentally friendly and non-toxic reagents, in contrast to other wet chemical synthesis methods, which employ noxious/toxic chemicals that can later be translated into the final products, therefore affecting the use of such NPs in pharmaceutical and other medical/biomedical applications. The advantages of green synthesis, besides the increased biocompatibility of the obtained NPs, are based on the control of the NPs morphology, lower costs and the fact that the enzymes and proteins found in the source materials are good reducing and capping agents [117]. In this respect, microbes (fungi, algae, bacteria), plant extracts of leaves, roots, fruits, or flowers (terpenoid, alkaloid, tannins, phenol, polyphenol, etc.) or various biological products (starch, egg protein, honey, agarose, pectin, etc.) are used [118,119,120]. Microbial synthesis of MONPs was shown to be an advantageous method due to its reduced toxicity compared to the typical high-pressure and chemical processes [121], but also because the microbes are not detrimentally affected by the synthesis conditions [122]. The green synthesis of MONPs using plant extracts is based on the fact that the plant extracts are employed in the bioreduction of metal ions and to synthesize and stabilize the NPs; similarly, the process is simple, fast and environmentally friendly. Overall, in green synthesis, the reaction rates are slower and only a limited variety of NP shape and size can be obtained. Due to the above-mentioned advantages, this synthesis method is widely used for the growth of MONPs for biomedical applications, targeting various NPs such as Ag [121,123], or other metal oxides, as described in recent works or reviews (ZnO [8,37,117], Fe_2_O_3_ [124,125], CeO_2_ [117,119], TiO_2_ [117,126,127], MgO [128], CuO [129], NiO [117,130], ZrO [131,132] or ZrO_2_ [133] and CdO [62,134]).

On a final note, for more details with respect to the synthesis approach and methodology (used precursors, additives, stabilizers, or reactions conditions), the following reviews are recommended for the synthesis of various MONPs, i.e., for ZnO [8,37,120], CeO_2_ [119,135,136,137], Fe_2_O_3_ [124], TiO_2_ [120,138], MgO [63,128,139,140], CuO [129,141], NiO [30,142,143] and CdO [144] NPs.

Additionally, in view of the biological effects (interactions with biofluids, cells, biomolecule, etc.) of such MONPs, these are influenced by a wide range of factors, such as NPs size, aggregation state, morphology and stability and, therefore, the synthesis methods are typically tailored towards achieving control over the NPs morphology, size and stability. For example, the physical and chemical properties of MONPs that have an impact on the interactions with cells are the (i) NPs morphology (shape, size), which controls aspects such as overcoming cell barrier, internalization and toxicity [145,146,147,148]; (ii) NPs surface area and surface energy, as this influences the number of active sites and can control reactivity [82,149]; (iii) crystal structure, which together with size, defects, media composition and aggregation, influences the dissolution of the metal ions, which can cause toxic effects [150,151]; (iv) surface chemistry, such as surface charge (zero-point of charge, acidity constant), dispersibility and aggregation, influence surface cascade reactions consequential for healing and subsequent biointegration [150,151,152,153,154,155,156,157]; (v) photocatalytic properties and chemical composition of the MONPs, as some nanoparticles can generate hydroxide or peroxide radicals and, furthermore, can (photo)release metal ions that may either promote adsorption reactions and/or facilitate favorable/unfavorable localized ‘-cidal’ effects [158,159,160].

### 2.2. Functionalization of Metal Oxide Nanoparticles for Biomedical Applications

The previous section discussed the various synthesis methods of MONPs, with an overview of ZnO, Fe_2_O_3_, CeO_2_, TiO_2_, MgO, NiO, ZrO and CdO NPs. While some of these NPs already present some biological effects in their bare nanoparticulate form, the performance and use of others type of NPs can be maximized by additional modifications. These modifications involve the surface functionalization of NPs such that they can elicit specific responses that may be biologically or chemically more favorable.

It should be noted that most of the synthesis processes result in hydrophobic NPs, as a result of synthesis conditions and due to the use of surfactants. This, in turn, limits the solubility of the NPs in aqueous or biological media [161]. There are many approaches for surface modifications and these include functionalization with drugs, polymers, biopolymers, inorganic materials, or bioconjugation (Figure 2). This is achieved by methods such as coating, conjugation strategies, in situ synthesis, self-assembly, surface encapsulation, or the synthesis of core-shell nanoparticles. After the surface modification, the functionalized nanoparticle is compatible with the biological environment, predominantly due to the hydrophilic nature of the coated shell [155,162].

With respect to drug functionalization, the key advantages of using MONPs are twofold. The first is connected to the possibility of localizing the drug to the target cell or area, which significantly increases the potency of the drug, while reducing the dosage and, thus, removing the issue of toxicity to the tissue. Secondly, having a surface coating (shell) on the MONPs can stabilize the nanoparticles, influence the size of the colloid particle and their bio-kinetics and distribution in the body, as well as diminishing their toxicity [155,161,163]. A wide range of specific drugs can be employed, such as anticancer, anticonvulsants, immunosuppressants, antibiotics, anti-inflammatory, antiviral, antifungal, or alternative, drugs. The drugs can either be covalently bound to the MONPs surface or via electrostatic interactions or via sequential functionalization [164] such that loading and release kinetics are governed by affinity to binding substrates and localized environments.

MONPs can be modified by polymers and/or biopolymers, which also contributes to nanoparticle stability in physiological conditions, increase their activity towards biological interactions, and can be further used to introduce more diverse functionalities [161,165,166]. Such polymer coatings can be achieved by either replacing an initial coating on the MONPs (e.g., ligands) or by directly coating the polymer. Typical examples of polymers used include poly(ethylene glycol), poly(lactic-co-glycolic acid), poly(vinyl alcohol), poly(lactic acids), poly(vinylpyrrolidone), poly(alkyl cyanoacrylates), poly(e-caprolactone), poly(methyl methacrylate), poly(ethyleneimine) and poly(dopamine) [155,161,167]. To further tackle the issue of toxicity of polymers at higher concentrations of longer treatment duration, an alternative is represented by using biopolymers such as peptides, proteins, dextran, chitosan, heparin, cellulose, lignin, etc. [155,161].

The use of inorganic moieties or materials such as surfactants, e.g., sodium dodecyl sulfate and sodium oleate, inorganic ligands such as carboxylates, silanes, phosphates and so on, or silica, is also widely employed for establishing a coating on the MONPs. For example, silica significantly increases NPs stability, biocompatibility and surface functionality with respect to biomedical applications, and is used for coating the surface of magnetic nanoparticles [161].

Another approach used for the functionalization of the MONPs is bioconjugation, which consists of the conjugation of NPs surfaces with biomolecules whose tailored properties evoke favorable interactions in the biological environment [168]. Often, linker molecules are necessary to obtain good adhesion and functionality of the immobilized biomolecules [169]. These conjugations enable the NPs to reach and effectively interact with site-specific cells [161,170].

Furthermore, the functionalization method also depends on the chemical nature and surface properties of the chosen nanoparticles, and thus there is no universal method valid for all MONPs. For detailed information with respect to specific functionalization approaches targeting the discussed MONPs of the present review, readers are referred to the following literature reports—ZnO [155,161], Fe_2_O_3_ [155,161,163,171], CeO_2_ [172], TiO_2_ [23,155,173,174], MgO [139,140] NPs, or recent functionalization approaches (NiO [175] NPs).

### 2.3. Characterization of Metal Oxide Nanoparticles for Biomedical Applications

The typical characterization techniques for MONPs also in view of targeting biomedical applications are based on evaluating the: (a) morphology and composition—scanning electron microscopy (SEM) and transmission electron microscopy (TEM), combined with energy dispersive X-ray (EDX) analysis; (b) crystallographic structure—X-ray diffraction analysis (XRD); (c) molecular groups and chemical bonding—Fourier-transform infrared spectroscopy (FTIR), or time-of-flight secondary ion mass spectrometry (ToF-SIMS); (d) NPs’ chemical and compositional properties—X-ray photoelectron spectroscopy (XPS); (e) synthesis mechanism—thermogravimetric analysis and differential thermal analysis (TGA–DTA); and (f) evaluation of the bandgap adsorption peak—UV-Vis spectroscopy. Moreover, other typical characterization techniques include the evaluation of the zeta potential, which is crucial in evaluating the effective electric charge of the nanoparticles (without or with further functionalization of the NPs). The different characterization techniques are also chosen as a function of the MONPs evaluated, due to the specifics of the metal oxide material and/or their further modifications.

#### 2.3.1. MONPs—Morphology Evaluation

From the above characterization methods, electron-microscopy techniques are crucial for the evaluation of the nanostructure and the NPs morphology with respect to particle size (mean and distribution), as well as providing detailed structural information at the atomic scale [176].

Representative TEM images of different MONPs obtained by various synthesis methods are shown in Figure 3. Namely, Figure 3a shows a representative high-resolution transmission electron microscopy (HRTEM) image of ZnO NPs obtained by a green synthesis using *E. prostrata* leaf extract as a capping agent—the NPs have an average size of 29 nm, ranging from 16 to 85 nm, showing also a triangular, radial, hexagonal, rod, or rectangular shape [177]. The SAED pattern is also included (selected area electron diffraction pattern) confirming the high crystallinity of the NPs. Figure 3b shows the HRTEM image demonstrating the formation of small-sized CeO spherical particles (diameter ~5 nm), obtained by a simple wet chemistry method [178,179]. Balaji et al. [180] have synthesized different sizes of biogenic ceria (CeO_2_) NPs, with diameters of 50, 20, 10 and 5 nm, by hydrothermal synthesis in the presence of *E. globulus* leaf extract—Figure 3c shows the HRTEM image of the CeO_2_ NPs with 20 nm diameter. Moreover, the authors confirmed the formation of CeO_2_ particles with a fringe space of 3.1 Å, also corroborated by the XRD data, demonstrating a (111) plane (3.24 Å) of CeO_2_ NPs [180]. Rufus et al. [181] have reported on the green synthesis of α-Fe_2_O_3_ NPs (environmentally benign, by the use of guava leaves, Psidium guajava) via a simple precipitation method. While SEM confirmed the quasi-spherical shape of the NPs, with diameters in the 20–48 nm range and an average diameter of 35 nm (weight of iron and oxygen from EDX was 62.55% and 37.45%), further TEM characterization corroborated the irregular shape of the NPs with an average size of ~38 nm (Figure 3c) and a rhombohedral structure (lattice fringe width of 0.27 nm corresponding to the (104) facets of the rhombohedral structure—Figure 3d).

TiO_2_ NPs obtained by thermal decomposition from titanium oxysulfate and urea (precursor ratio 1:0.4) are shown in Figure 3e [94], and the diameter of the NPs can be controlled by the urea amount—in this case, the NPs had diameters in the 20–40 nm range. MgO NPs obtained by a sol-gel method with the addition of a surfactant (in order to prevent agglomeration) are shown in Figure 3f, having some light agglomeration but with a narrow distribution of the particle size (average size 15.7 nm) [70]. Hydrothermally grown NiO NPs with an average size of 29 nm are shown in Figure 3h (authors confirmed the NPs size by STM measurements) [182]. Green-synthesized ZrO NPs, using the leaves of *L. speciosa*., are shown in Figure 3i, with an average particle size of 56.8 nm with a tetragonal morphology (this could be due to the green synthesis, as the biomolecules are capping the NPs) [131]. CdO NPs obtained by a recently reported annealing of polyvinyl alcohol and para-aminobenzoic acid complexes (from an aqueous solution containing metal chloride as a precursor) [183], with an average diameter of 58 nm, are shown in Figure 3j.

Figure 3 clearly shows the variety in NPs shape and size, consequently dependent on the synthesis procedure, together with the progress made into avoiding NP agglomeration via the use of additional capping or stabilizing agents (e.g., plant-based extracts to further ensure the NPs eco-friendliness and biocompatibility, and, thus, their use in biomedical application).

#### 2.3.2. MONPs—Crystallographic Structure Evaluation

X-ray diffraction represents a widely used characterization technique in material science in order to determine the crystallographic structure of the materials, and it is also a crucial evaluation tool for MONPs. Especially as some synthesis techniques require an annealing/thermal treatment step, known also as post-synthesis, to crystallize the NPs (e.g., chemical precipitation), while other methods result directly in the formation of crystalline NPs (e.g., hydrothermal synthesis).

XRD is especially needed in the case of synthesis methods where the effect of the thermal treatment on the crystallinity of the NPs has to be evaluated. For example, for the green synthesis of ZnO NPs in the presence of cyanobacterium from *A. Platensis* [184], a wurtzite structure was confirmed (Figure 4a (peaks at 2 theta degree 31.7°, 34.5°, 36.1°, 47.4°, 56.3°, 63.1° and 67.9°, which matched to the (100), (002), (101), (102), (110), (103) and (112) planes, respectively), with an average crystal size of ≈45 nm (computed from the XRD analysis by the Debye–Scherrer equation)). In the green synthesis of CeO_2_ NPs in the presence of *Prosopis farcta* leaf extract [185], the authors evaluated the impact of the temperature on the synthesis and determined a fluorite cubic structure of the CeO_2_ NPs; a similar structure was reported when synthesis was performed with *Eleagnus angustifolia* leaves [186].

Overall, XRD represents a necessary characterization technique to evaluate the crystallinity, lattice parameters, Miller indices and crystallite size for most of the studied MONPs (e.g., ZnO [177,184,187], Fe_2_O_3_ [84]_,_ MgO [188], NiO [182], ZrO [131,132] and CdO [183]), irrespective of the synthesis method.

#### 2.3.3. MONPs—Chemical and Compositional Evaluation

The chemical and compositional structure of the MONPs, without or with further functionalization, can be evaluated by several complementary techniques. FTIR can be used to identify the chemical bonds and characteristic functional groups, especially in view of functionalized MONPs or biomedical composites [189,190]. For example, for the green-synthesized ZnO NPs (cyanobacterium from *A. Platensis* [184] of Figure 4a), the functional groups and chemical structures can be determined by FTIR (Figure 4b). Peaks are observed and their assignments were: 3415 cm^−1^—N–H overlap with a stretching O–H band, 3000 cm^−1^—stretching CH_2_ of asymmetric and symmetric carbohydrates and/or lipids, 1600 cm^−1^—stretching C=O vibration of proteins or remaining acetate, 1410 cm^−1^—C–N stretching bond of amino acid, 1341 cm^−1^—vibration bending of the C–H (absorption wave of CH_2_ or CH_3_ of proteins), 1025 cm^−1^—C–O–C ether of polysaccharides, 676 cm^−1^—C=C bonds and 503 cm^−1^—Zn–O absorption band [184]. In addition to confirming the formation of the ZnO nanoparticles, data show the role of organic substances present in the *A. platensis* extract in the reduction, capping and stabilization of the biosynthesized ZnO NPs [184].

**Figure 4 jfb-13-00274-f004:**
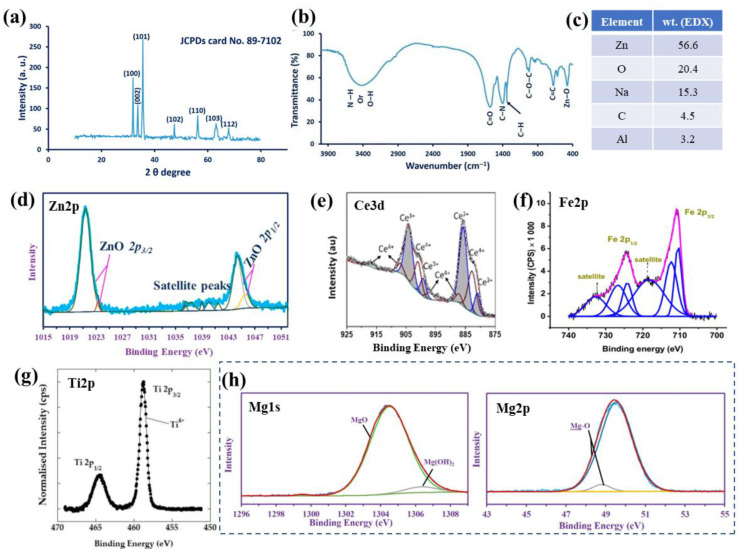
Green-synthesized ZnO NPs in the presence of *A. platensis* (cyanobacterium): (**a**) XRD patterns, (**b**) FTIR spectrum, (**c**) weight percentage from EDX (data from ref. [184]), and (**d**) high-resolution XPS peak of Zn2p (**a**,**b**,**d**: reprinted from ref. [184]). High-resolution XPS spectra for (**e**) Ce3d of CeO NPs (reprinted with permission from ref. [178]. Copyright 2021 Elsevier), (**f**) Fe3d in Fe_2_O_3_ NPs (solvothermal synthesis in the presence of double capping agents) (reprinted from ref. [191]), (**g**) Ti2p in TiO_2_ NPs synthesized by microwave-assisted method (reprinted from ref. [192]), and (**h**) Mg1s and Mg2p in MgO NPs (biosynthesis in the presence of metabolites from *Penicillium chrysogenum*) (reprinted from ref. [193]).

EDX can be used to evaluate the elemental composition of the NPs, and EDX coupled with TEM provides local chemical composition and mapping of the NPs. Similarly, for the ZnO NPs synthesized via a green route (cyanobacterium from *A. Platensis* [184]) of Figure 4a,b, EDX was employed to evaluate the quantitative elemental structure and confirmed the presence of Zn, O, Na, C and Al with weight percentages of 56.6, 20.4, 15.3, 4.5 and 3.2%, respectively (Figure 4c); thus, confirming the ZnO NPs formation through the use of the metabolites in the *A. platensis* filtrate [184].

XPS provides information on the chemical and compositional properties of surfaces and is especially employed to confirm the composition of the NPs, as well as their further modification/functionalization. Typically, the adventitious carbon peak is used to calibrate the measured spectra, nevertheless, recent works have shown the limitations and how to reliably determine the chemical states [194,195,196] and to accurately fit the peaks of interest [196,197]. Using, as an example, the green-synthesized ZnO NPs [184], the authors confirmed the presence of Zn(II) [184,198]. Namely, Figure 4d shows the high-resolution Zn 2p peak and its deconvolution into the doublet with Zn 2p3/2 at 1021.4 eV and Zn 2p1/2 at 1044.2 eV, and the doublet with the 2p3/2 at 1023.25 eV and 2p1/2 at 1045.55 eV (with satellite peaks at 1036.25, 1037.3, 1039, 1040.05, and 1041.65 eV, verifying the oxide species) [184,198]. Additionally, for MONPs, deconvolution of the O1s peak can also be performed to verify the presence of the metal oxide, and in the case of biosynthesis or further functionalization peak fitting of the C1s peak is also crucial. For example, for the biosynthesized ZnO NPs, the authors verified the hydrocarbon composition produced in their reaction medium with peak fitting of the C1s (five peaks at 284.48, 285.75, 287.9, 287.05 and 288.9 eV for C(H, C), C–N, C–O, C=O and C–O–C [184]), and the oxide structure by fitting the O1s peak (overlap of the O in ZnO with that in NaO; Na KL1 at 536.75 and O(C, H), O=C and C–O–C at 531, 532.2 and 535.3 eV, respectively) [184].

Figure 4e–h further show the typical high-resolution XPS peak of the corresponding metal element from the CeO, Fe_2_O_3_, TiO_2_ and MgO NPs. From these MONPs, the spectra of Ce 3d and Fe 2p are quite complex. Namely, Figure 4e presents the Ce 3d peak of CeO NPs [178] with the 3d doublet (3d5/2 and 3d3/2) indicating the Ce^3+^ and Ce^4+^ states with 60% of the cerium being present as Ce^3+^ (Ce^3+^ peaks of two spin-orbit features: ~880.6, 885.5, 898.8 and 903.7 eV and Ce^4+^ peaks of three spin-orbit features: 882.5, 887.1, 897.2, 900.7, 906.8 and 916.3 eV). The XPS spectra clearly show the presence of both chemical states, and the presence of the peak at 916.3 eV enables the clear differentiation between the Ce^3+^ and Ce^4+^ states, as this peak arises only for the Ce^4+^ state [199,200,201]. In addition, the authors [178] evaluated the O1s spectra and reported a typical asymmetry as a result of the O^2−^ ions from different chemical environments (e.g., oxygen bound to Ce^4+^, Ce^3+^ and H^+^) thus further corroborating the dual oxidation states. In the case of Fe 2p, Figure 4f shows a typical Fe 2p peak and its deconvolution, for monodisperse magnetic γ-Fe_2_O_3_ nanoparticles obtained by the solvothermal method (with double capping agents) [191]. The differentiation between iron oxides is possible as the spectrum of Figure 4f is typical to the Fe^3+^ state of the γ-Fe_2_O_3_ NPs, consistent with literature data [202,203], that is with Fe 2p3/2 at 710.85 eV and Fe 2p1/2 at 724.42 eV, with their corresponding satellites at 732.78 and 718.44 eV [191].

Typically, the Ti 2p peak corresponding to TiO_2_ (NPs, or other nanomorphologies) is more straightforward with respect to the peak shape and chemical state, due to the presence of only one oxidation state of titanium. For example, Figure 4g shows a typical Ti 2p spectrum of TiO_2_ NPs, obtained, in this case, by a microwave-assisted method [192], and confirms the presence of Ti 2p3/2 and Ti 2p1/2 at 458.8 and 464.5 eV, respectively (attributed to Ti^4+^ of anatase TiO_2_ [204,205]). The authors also evaluated the O1s peak, which was deconvoluted into three peaks including 530.1 eV—assigned to oxygen bonded to titanium (O-Ti), 531.6 eV—oxygen bound to carbon (impurities from the synthesis, i.e., urea or acetylacetone) and 532.8 eV—adsorbed oxygen (O-H bonds of chemisorbed water) [192]. Similarly, in the case of MgO NPs, for example, obtained by (biosynthesis harnessing the metabolites secreted by *Penicillium chrysogenum* [193], the presence of MgO as the main species was confirmed. Briefly, the Mg1s spectrum was deconvoluted into MgO at 1304.44 eV (94.74%) and Mg(OH)_2_ at 1306.28 eV (5.26%), and the Mg 2p confirmed this point with Mg-O bonds at 49.49 eV (95.89%) and Mg-OH bonds at 48.74 eV (4.11), as shown in Figure 4h—consistent with literature data [205,206] and further confirmed with the deconvolution of the Mg 2s spectra [193].

For NiO NPs, the typical Ni 2p peaks, i.e., Ni 2p3/2 and Ni 2p1/2, can be observed at binding energies of 853.7–855.42 eV with a split spin-orbit of 17.3 eV [205], and, additionally, with satellite peaks at 861.5, 867.16 and 879.3 eV [207]. In the case of CdO NPs, the Cd 3d peaks attributed to the cadmium oxide are expected at 405 eV (Cd 3d5/2) and 412 eV (3d3/2) [205,208]. 

Characterization techniques for establishing the chemical and compositional structure of MONPs are crucial for linking the structure of the NPs with their biological effects, and, in addition, prove necessary when the NPs are loaded with active molecules or have functionalized surfaces [209]. Even when used independently as ‘active’ agents in various composites targeting specific effects such as in the case of drug-loading/release, linking process and subsequent interactions, characterization techniques offer insights into interaction chemistry/behavior, etc.

## 3. Biological Effects

### 3.1. Pro-Regenerative Potential

Over the last few years, in the field of biomedical research, nanotechnology has offered numerous promising approaches for increasing the transition of regenerative medicine from research to clinical practice [24]. Amongst the variety of materials used in nanotechnology, NPs, especially MONPs, are a widely spread class of materials with unique physical and chemical properties that possess numerous advantages and multiple applications in the biological and biomedical fields [210] (Figure 5). One such application is in wound healing. Trauma, distinct skin conditions, burns, or removal of the skin due to surgical procedures, resulting in superficial or deep wounds, which can be prone to pathogenic colonization and further complications if not protected and treated properly [7]. In this context, suitable wound dressing materials that possess antibacterial properties and the ability to promote wound healing are necessary [211].

Taking this aspect into account, several wound dressings loaded with MONPs have proved able to decrease the infection and contraction time of the wound, without any significant side effects [212]. One such example of MONPs is ZnO NPs, which have been used with success in numerous wound dressings due to their strong antibacterial activity and stimulating effect on epithelial cells [213]. Raguvaran et al. [214] loaded ZnO NPs onto sodium alginate-gum acacia hydrogels (SAGA-ZnO NPs) and observed that, at low concentrations, these MONPs exerted wound-healing effects on sheep fibroblasts, whereas high concentrations proved to exhibit a cytotoxic effect. Moreover, the loaded hydrogel reduced the inherent toxic effect of the ZnO NPs, while keeping the antibacterial and healing properties of the NPs.

It is a well-known fact that wound healing is a complex process in which the presence of oxidative stress due to an over-production of ROS can lead to injured cells and tissues [215]. Keeping this in mind, numerous in vitro and in vivo studies focused on the suitability of MONPs for skin-wound repair and regeneration through the inhibition of ROS generation. For example, Davan et al. [216] observed in a rat model that spherical shape CeO_2_ NPs (with a size of 160 nm) were capable of enhancing the wound closure rate and collagen deposition, without scarring tissue. Similar results were observed in wound dressings loaded with CeO_2_ NPs; for example, Naseri-Nosar et al. [217] loaded poly(ε-caprolactone) (PCL)/gelatin films with CeO_2_ NPs and the results suggested that the film containing 1.5% CeO_2_ NPs is favorable in terms of L929 cells proliferation. In another study, Wu et al. [218] designed tissue adhesives using assembled ultra-small CeO_2_ NPs onto the surface of mesoporous silica NPs and the in vitro results showed the ability of the newly developed product to impair the exacerbation of ROS-mediated side effects and promote the wound healing process. Moreover, the in vivo results indicated a significantly low ROS level and a reduced local inflammatory activity, coupled with an improved wound healing and lack of scar tissue.

Another type of MONPs used for wound dressing studies are Fe_2_O_3_ NPs. One such study by Pai et al. [219] demonstrated that a composite thin film of poly(ε-caprolactone) -Fe_2_O_3_ NPs exhibited a strong antibacterial activity and promoted NIH 3T3 mouse fibroblasts proliferation. Similarly, Grumezescu et al. [220] prepared an absorbable wound dressing based on anionic polymers such as sodium alginate and carboxymethylcellulose and Fe_2_O_3_ NPs, and the results showed low cytotoxicity to human progenitor cells coupled with a powerful antibacterial activity. Another study showed the promising potential for wound healing of a silk fibroin-Fe_2_O_3_ NPs dressing that is biocompatible with human adipose stem cells (ASCs) [221]. Anghel et al. [222] developed a wound dressing coated with a nanofluid containing Fe_2_O_3_ NPs and two natural microbicidal compounds and observed that the coating exhibited anti-adherence and anti-biofilm properties against *Pseudomonas aeruginosa* (*P. aeruginosa*) and *Staphylococcus aureus* (*S. aureus*). Moreover, Fe_2_O_3_ NPs possess unique magnetic characteristics which can be used in order to achieve an accelerated wound-healing process. For example, Wu et al. [223] functionalized Fe_2_O_3_ NPs with basic fibroblast growth factor (bFGF) and reported an increased cell proliferation and macrophage polarization towards a pro-healing M2 phenotype. In a rat model, the administration of Fe_2_O_3_ NPs-loaded mesenchymal stem cells (MSCs) and their magnetically enhanced migration to the injury site improved skin regeneration and enhanced the anti-inflammatory effects and angiogenic process compared with only the injected MSCs [224].

In addition to their potential as wound dressings, MONPs (ZnO, CeO_2_ and Fe_2_O_3_) have been used in bone-regeneration applications due to their ability to augment the bone-healing process. For example, in a study by Tang et al. [210], *Scutellaria baicalensis* (SB)-ZnO NPs, i.e., with the addition of the Chinese herb *Scutellaria baicalensis* (generally used to treat bone and joint ailments), were investigated for their effects on osteoblast differentiation and osteoclast formation. The reported results indicated the ability of the SB-ZnO NPs to improve bone regeneration via osteoblast proliferation and differentiation enhancement and inhibition of osteoclast formation. Khader and Arinzeh [225] incorporated ZnO NPs in a PCL scaffold and the in vitro results suggested that the slow release of ZnO NPs from the structure of the composite benefited both the osteogenic and chondrogenic differentiation of MSCs. Garino et al. [226] evaluated the behavior of ZnO nanocrystals (NCs), with a diameter of 20 nm and partially chemically functionalized by anchoring amino-propyl groups, in terms of biocompatibility, cell proliferation and differentiation—it was suggested that the proposed NCs were capable of promoting bone tissue proliferation even at high concentrations. In another study, Zhou et al. [227] evaluated the effects of CeO_2_ NPs on the proliferation, differentiation and mineralization of primary osteoblasts and the results indicated that the biological activity of bone cells is size, concentration- and exposure time-dependent, with positive results at higher concentrations and with smaller size nanoparticles. Moreover, Yuan et al. [228] demonstrated that CeO_2_ NPs (average diameter 17 nm) are capable of inhibiting osteoclast formation and activity at high concentrations through the over-production of ROS. Similar results were reported by Wei et al. [178], where 5 nm CeO_2_ NPs were observed to enhance MSCs proliferation, osteogenic differentiation and mineralization. Li et al. [229] deposited CeO_2_ NPs on a titanium surface and investigated the underlying mechanism of new bone formation both in vitro and in vivo. The results showed that the prepared oxide NPs found in a mixed Ce^3+^/Ce^4+^ valence state promoted the new bone formation and mineralization even in the absence of specific osteogenic agents. Similar results were observed in another study [15], where in the absence of osteogenic agents, glass foam-based scaffolds coated with CeO_2_ NPs were able to enhance the collagen production and osteogenic differentiation of human mesenchymal stem cells (HMSCs), in comparison to CeO_2_ NPs-free scaffolds. In addition, Singh et al. [230] developed PCL nanofiber scaffolds functionalized with Fe_2_O_3_ NPs and observed improved cell adhesion and osteogenic activity of osteoblasts; while Cojocaru et al. [231] reported improved biocompatibility and bone cell proliferation for the newly developed biodegradable composite based on chitosan, calcium phosphate, hyaluronic acid and Fe_2_O_3_ NPs. Similarly, Lee et al. [232] loaded a nanoscaffold based on halloysite nanotubes with Fe_2_O_3_ NPs and observed that due to the osteoinductive abilities of the Fe_2_O_3_ NPs, the developed nanoscaffold was able to elicit an improved osteogenic differentiation of human adipose tissue-derived mesenchymal stem cells (hADMSCs) through the enhancement of osteoblast formation. Moreover, Zeng et al. [233] fabricated magnetic biomimetic hydroxyapatite (HA) scaffolds immersed in Fe_2_O_3_ NPs solutions and observed that, after cell proliferation, the murine pre-osteoblast MC3T3-E1 cell line and the rat osteosarcoma ROS 17/2.8 cell line experienced a promotion of the proliferative and differentiation processes. Similarly, in a study by Tanasa et al. [234], the presence of an applied magnetic field in scaffolds based on silk fibroin, poly(2-hydroxyethyl methacrylate) and Fe_2_O_3_ NPs (7 nm) led to an improvement in the proliferative state and differentiation capacity of the MC3T3-E1 pre-osteoblast cells.

However, despite the on-going progress made in this field, there are still many challenges that need to be overcome in order to obtain a successful transition from research to clinical practices; thus, further studies regarding the physicochemical characterization and in vitro and in vivo cytotoxic potentials of the MONPs are still required.

### 3.2. Antitumor Effect

#### 3.2.1. General Considerations

Cancer, a heterogeneous disease, which affects billions of people and is considered to be one of the main causes of mortality worldwide, represents a serious health problem. With the recent World Cancer Report by the World Health Organization stating that in 2020 the incidence of cancer increased to 19.3 million from 18.1 million in 2018, the growth trend is bound to continue, reaching up to almost 28.4 million cases per year in 2040 [235]. According to the National Cancer Institute (National Institute of Health), patients diagnosed with cancer are currently presented with several treatment options, which may include surgery, radiation therapy, hormone therapy, targeted therapy, biomarker testing, stem cell transplantation, etc. However, each of these therapies possesses the potential to impact the patients’ life quality, especially radiation and chemotherapy, which can cause side effects due to their difficulties in differentiating between cancer and normal cells, resulting in systematic toxicity [236]. The surgical approach might appear as a better option, but it has its limitations too, namely in the form of post-surgical scars and the inability to remove all of the tumoral mass, therefore requiring additional side therapy such as radiation, chemotherapy, or, in extreme cases, both. In this context, targeted therapies minimize the side effects whilst improving patient care. New approaches for cancer treatment continue to be studied and developed, and one such strategy includes the use of MONPs against tumor development and progression, due to their intrinsic antitumor effects [10]. The exhibited anticancer activity of MONPs is related to their unique physicochemical properties, which are either related to their intrinsic features, such as their antioxidant action or depend on activities based on the application of external stimuli [10]. In addition, MONPs possess the ability to transport anticancer drugs to a specific tumor location. This specific targeting is achieved by using either an active or passive process. Passive targeting is mainly based on the enhanced permeability and retention effect, meaning that the leaky vasculature found in tumoral tissue allows MONPs to diffuse rapidly and kill cells [237]. However, some adverse effects are associated with drug delivery via passive targeting; for example, the leaky vasculature found in the tumoral mass can also be present in inflamed tissue; therefore, rendering the targeted drug delivery less than ideal due to the lack of precision. Conversely, drug delivery via active processes can reduce the side effects caused by passive targeting, due to the fact that the NPs are functionalized and directed specifically against the cancer cells. Thus, through biomolecule or ligand binding to the surface of the NPs, the targeted delivery of anticancer agents to tumor cells instead of normal ones, can be improved [10].

#### 3.2.2. Applications of ZnO NPs in Cancer Therapy

In anticancer therapy, MONPs are used experimentally to kill tumor cells both in vitro and in vivo. Amongst several biomedical applications, the use of ZnO NPs in cancer therapy has been well explored. The antitumor activity of ZnO NPs stems from both the ability to generate ROS and their electrostatic properties [37]. The selective toxicity of ZnO NPs against cancer cells has been demonstrated in an in vitro study through the use of co-cultured C2C12 myoblastoma cells and 3T3-L1 adipocytes. The results showed that the levels of ROS and p53, bax/bcl-2 ratio and caspase (CASP)-3 enzyme activity were increased in co-cultured C2C12 cells in comparison with the 3T3-L1 adipocytes, suggesting that the ZnO NPs selectively induced apoptosis in the C2C12 cancer cell [238]. In addition, Wahab et al. [239] demonstrated the specificity of ZnO NPs by investigating their toxic effects against malignant T98G human gliomas and KB epidermoids in comparison to non-tumoral HEK kidney cells. The ZnO NPs were found to exhibit a strong cytotoxic effect against the T98G cancer cells, a moderate effect against the KB cells and an insignificant effect on the healthy kidney cells. Similarly, Premanathan et al. [240] reported that ZnO NPs are capable of inhibiting the proliferation of human myeloblastic leukemia cells in comparison to normal peripheral blood mononuclear cells. In addition, Pandurangan et al. [241] investigated the cytotoxicity of ZnO NPs in human cervical carcinoma cells and it was demonstrated that the cancer cells’ viability was significantly reduced, therefore suggesting the possible cytotoxic effect of ZnO NPs through the overproduction of ROS. Furthermore, ZnO NPs (80, 150, 260 and 400 nm in average diameter) were reported to exhibit a cytotoxic effect on ovarian cancer cells, through the induction of acute oxidative and proteotoxic stress, which led to cell death via apoptosis [242]. In another study, Shahnaz et al. [243] observed that ZnO NPs (12–26 nm) were able to induce cytotoxic effects on the HCT-116 colon cancer cell line in comparison to the Vero cell line. In addition to studies with ZnO NPs as standalone agents, numerous studies focused on modified ZnO NPs due to their improved stability and increased selectivity for specific cells. Results showed that surface modifications using Triton-X, polyethylene glycol (PEG), or hyaluronan did not affect the antitumor activity of the ZnO NPs but did improve their safety towards normal cells due to their biocompatible coating [244,245,246]. In other studies, ZnO NPs have been coated with doxorubicin (DOX), cisplatin and paclitaxel (PTX) and the results indicated that their cytotoxic effect increased significantly in combination with this type of MONPs [53,239]. Wu and Zhang [247] investigated the anticancer effect of both chitosan-coated and uncoated ZnO NPs in HeLa cells exposed to different concentrations and the results obtained showed that both coated and uncoated NPs exhibited reduced cytotoxicity when exposed to smaller concentrations, whereas the chitosan-coated positively charged ZnO NPs caused enhanced cytotoxicity at higher concentrations, possibly through the increased cellular internalization and subsequent ROS production, which caused cellular death by apoptosis. Given the fact that ZnO NPs possess inherent antitumor properties, researchers have used them as drug-delivery platforms for several active biomolecules and drugs. ZnO NPs-based drug-delivery systems (DDS) possess several advantages, such as (i) low risk of systemic toxicity due to the inhibition of a premature release of the loaded drug; (ii) they offer the loaded drugs an increased aqueous solubility and improved hydrophobicity; (iii) they increase the drugs’ efficiency by transporting them to the targeted cells/tissues/organs via an active process; and (iv) show a low risk of cytotoxic effects towards normal or healthy cells/tissues/organs. 

Presently, only four types of DDS based on ZnO NPs are mainly adopted: (i) mesoporous silica nanoparticles (MSN)-based DDS; (ii) porous ZnO NPs where the active drugs are loaded inside the pores; (iii) ZnO NPs/polymer core-shell nanocomposites where the drugs are loaded into the hydrophobic shell; and (iv) ZnO NPs/drug complex [248]. Zhang et al. [249] developed a multifunctional MSN-based charge reversal and ZnO quantum dots (QDs) targeted drug-delivery system for combined cancer therapy. In order for the MSNs to be able to escape easily and more rapidly from endosomes, they were functionalized with cell-penetrating deca-lysine peptide, while the positively charged ZnO QDs were used to cap the DOX-loaded MSN pores through electrostatic interactions. The results indicated a synergistic anticancer effect in Hep G2 cells through the targeted release of DOX from the uncapped MSN pores into the cytosol. Similarly, Cai et al. [250] designed ZnO QDs functionalized with hyaluronic acid (HA) for pH-responsive delivery of DOX in A549 cells. The mechanism behind this drug-delivery platform is based on the recognition of highly expressed CD44+ cells and the release of drugs through the rupture of the metal-DOX complex due to the dissolution of ZnO NPs in the acidic intracellular compartment. It was demonstrated that the HA-functionalized ZnO QDs-DOX exhibited higher cytotoxicity compared to the non-targeted ZnO QDs-DOX due to the increased intracellular uptake. In another study, Wang et al. [251] reported the successful delivery of certain immune-stimulating agents such as ovalbumin and polyinosinic-polycytidylic acid, with the help of hollow ZnO nanospheres for cancer immunotherapy, showing that the combination of the drug-loaded NPs significantly reduced the tumor growth and metastasis to the inguinal lymph node in the E.G7-OVA cell line. Akbarian et al. [252] developed a DDS for paclitaxel (PTX) based on chitosan-coated ZnO NPs and observed that the PTX-loaded ZnO-chitosan NPs exhibited a cytotoxic effect on MCF-7 cells, and a minimal effect on normal fibroblasts, suggesting that these newly developed ZnO-chitosan NPs could be used as a promising drug-delivery platform for PTX. A biocompatible co-polymer encapsulated ZnO NPs with an interior hydrophobic core designed for efficient encapsulation of curcumin proved to exhibit a higher cytotoxic effect against human gastric cancer cells in comparison to nanocurcumin [253], while a ZnO/ferulic acid stable nanohybrid showed a synergistic antitumor potential in human carcinoma Huh-7 and HepG2 cell lines through the induction of ROS, oxidative stress and DNA damage, followed by cycle arrest in the S phase and intrinsic apoptosis pathways. Moreover, the in vivo results indicated a significant reduction in the number of hepatic nodules and tumor-associated toxicity in hepatocellular carcinoma (HCC) bearing mice [254]. 

Abbasian et al. [255] synthesized cationic cellulose based ZnO nanocomposites and investigated the targeted and pH-responsive delivery of methotrexate (MTX) into MCF-7 breast cancer cells. The anticancer agent MTX was loaded into the newly developed nanocarriers via electrostatic interactions generated between the drug’s carboxyl groups and the cationic moiety of the NPs and by the formation of ZnO complexes at the chelating sites of MTX. The results showed a higher cytotoxicity against the MCF-7 cells in comparison to the free MTX, probability due to its increased intracellular uptake. Table 1 summarises additional in vitro and in vivo studies focused on investigating the antitumor potential of ZnO NPs either as stand-alone agents or as drug-delivery platforms.

#### 3.2.3. Applications of CeO_2_ NPs in Cancer Therapy

Cerium oxide NPs are a novel and very interesting compound, which are currently pursued in various in vitro and in vivo studies for their potential use in cancer treatment (Table 2). Being originally investigated for their antioxidant activity and ability to protect normal cells/tissues from radiation-induced damage associated with cancer therapy in the intestine [272], head and neck [273], breasts [274] and lungs [275], the use of CeO_2_ NPs has expanded beyond the prevention of adverse side effects of other cancer treatments. For example, data found in the literature indicates the inherent toxicity of CeO_2_ NPs towards various cancer cells such as pancreatic carcinoma cells [276], hepatocellular carcinoma cells [277], epithelial cancer cells [278], melanoma cells [279], ovarian cancer cells [280], etc. Taking this into account, the use of CeO_2_ NPs in cancer therapy is ever-growing, with the NPs being used both as the primary treatment and as an adjuvant treatment for the already in-use therapies [281]. In 2006, Lin et al. [282] evaluated the antitumor activity of different concentrations of CeO_2_ NPs in A549 human lung cancer cells and observed a dose- and time-dependent cytotoxicity towards the tumor cells through the induction of ROS and implicitly oxidative stress. Similarly, the inherent toxic effect of CeO_2_ NPs was reported on human colon cancer cells (HCT-15) in a dose- and time-dependent manner [283].

Kumari et al. [285] investigated the cytotoxic effect of CeO_2_ either as NPs or as microparticles for 24 h in human neuroblastoma cells and the results indicated that the tumor cells treated with NPs showed a higher production of ROS and subsequently a higher cytotoxic effect in comparison to the microparticle structure. Another cell line sensitive to CeO_2_ NPs toxicity is WEH1164, which was demonstrated by Nourmohammadi et al. [286] to exhibit a dose-dependent sensitivity. Furthermore, Renu et al. [284] prepared cerium oxide NPs via two different methods in order to obtain ceric oxide NPs with a +3 oxidation state (hydrolysis) and cerous oxide NPs with a +4 oxidation state (hydrothermal) and the results demonstrated that the hydrothermal NPs possessed a higher cytotoxicity towards prostate cancer cells compared to the hydrolysis NPs, mainly due to their increased cellular uptake. However, when compared to normal mouse fibroblast cell line L929, no toxic effects could be observed. Furthermore, Giri et al. [280] investigated the in vivo effect of CeO_2_ NPs on A2780 xenograft tumor mice models and after intraperitoneal administration of NPs for every third day up to 30 days. They observed that the tumor weight and the abdominal circumference in the treated mice were significantly reduced compared to the untreated mice. These results suggested that such NPs possess the ability to inhibit metastasis and the angiogenic process in ovarian cancer cells and implicitly reduce ovarian tumor growth. In another in vivo study, Hijaz et al. [43] evaluated the anticancer effect of CeO_2_ NPs modified with folic acid in A2780 xenograft tumor mice models and it was reported that the folic acid-tagged NPs were more efficient in attenuating the tumor growth in the treated mice compared to the untreated animals.

Recently, CeO_2_ NPs have also been widely used as effective drug-delivery platforms for various active drugs. This drug-delivery property of CeO_2_ NPs is based on their inherent cytotoxicity towards tumor cells, exhibiting a synergistic anticancer effect. In 2014, Muhammad et al. [287] designed a redox-responsive CeO_2_ NPs capped MSN-camptothecin delivery platform for the active transport of an anticancer drug into the human pancreatic cancer cells and reported that such a drug-delivery platform was capable of inducing a dose- and time-dependent cytotoxic effect on the tumor pancreatic cells, mainly due to its active intracellular uptake and dissolution of the NPs lid in the highly acidic microenvironment, which led to the release of the encapsulated drug. Moreover, Li et al. [44] developed a CeO_2_ NPs-based drug-delivery platform by conjugating a photosensitizer, chlorin e6 and folic acid on polyethylenimine-PEGylation CeO_2_ NPs for a targeted photodynamic treatment against drug-resistant human breast cancer cells and xenograft murine models. Under near-infrared irradiation (NIR), the newly developed drug-delivery system was capable of generating ROS, leading to a reduction in the P-glycoprotein expression, and an increase in the lysosomal membrane permeabilization, which in turn results in cytotoxic effects towards breast cancer cells even at lower doses. Furthermore, the in vivo results revealed that in the presence of the irradiation procedure, the mice treated with the CeO_2_ NPs system showed a visible reduction in tumor growth up to almost 98%. In another study, a CeO_2_ NPs-DOX drug-delivery system exhibited a higher degree of apoptosis and inhibition of the cell proliferative rates compared to free DOX in human ovarian cancer cells [288]. Doxorubicin was used as a loading agent in another study by Zhang et al. [290] where a multifunctional and pH/GSH (glutathione) dual-responsive drug-delivery system using porous CeO_2_ NPs was developed in order to target human liver cancer cells. The authors reported a synergistic anticancer effect against the tumoral liver cells, probably due to the low intracellular pH and high GSH levels inside the lysosomes present in the cancer cells. Sulthana et al. [289] designed polyacrylic acid (PAA)-coated CeO_2_ NPs loaded with a combination of drugs (Hsp90 inhibitor, ganatespib and DOX) for the treatment of non-small-cell lung cancer. They observed that this delivery platform led to a reduction in cell viability to almost 80% in comparison to the single drug-delivery system. In addition, Kalashnikova et al. [291] explored the anticancer effects of dextran-coated CeO_2_ NPs loaded with curcumin in human childhood neuroblastoma and reported that the newly developed DDS was capable of inducing a significant toxic effect on the neuroblastoma cells without affecting the healthy cells.

#### 3.2.4. Applications of Fe_2_O_3_ NPs in Cancer Therapy

Due to their non-toxic, biodegradable and cheap nature, Fe_2_O_3_ NPs have been extensively studied as potential candidates for different cancer therapies [292,293]. Fe_2_O_3_ NPs are magnetic biomaterials that can be directed and concentrated by external magnetic fields, e.g., NIR or oscillating magnetic fields (MF), and removed easily once the treatment is brought to completion [294]. Data found in the literature indicate that Fe_2_O_3_ NPs are capable of killing tumor cells without affecting normal healthy tissue due to the increased in vivo sensitivity of tumors to heat damage. This allows the use of a specific cancer therapy called hyperthermia, where magnetic NPs can target tumors in a heat-specific manner through the alternation of fields, hysteresis and frictional heating [295]. Anticancer hyperthermia therapy implies the use of heat temperatures above 40 °C. For example, Hilger et al. [296] injected supermagnetic NPs into immunodeficient mice models with implanted breast adenocarcinoma cells and observed an increase in temperature within the tumor region of up to 73 °C, but only after applying a 400 kHz magnetic field. Therefore, by employing the use of Fe_2_O_3_ for cancer therapy, the risk of damaging healthy tissue is significantly reduced, while the selectivity for cancer cells is greatly improved [10]. Moreover, magnetic Fe_2_O_3_ NPs allow for differential functionalization or surface loading, which can be especially useful for magnetically assisted drug-delivery treatments. Therefore, this type of NPs can be coupled with antitumor agents, either by covalent binding or through co-encapsulation in various polymeric matrices. To date, several active molecules, such as DOX and PTX have been loaded and tested as potential anticancer agents [297,298]. For example, Plichta et al. [299] reported a reduction in human glioblastoma cells’ viability when treated with magnetic γ-Fe_2_O_3_ NPs conjugated with DOX at low concentrations, while in A549 lung cancer cells, PEG-functionalized γ-Fe_2_O_3_ NPs conjugated with DOX were capable of inducing an increase in the viability rate, possible due to the insufficient release of DOX from the system. However, when an alternating magnetic field (AMF) was employed, the NPs exhibited excellent thermal effects that favored the release of DOX from the delivery platform and implicitly the death of the lung cancer cells [300]. Likewise, Lungu et al. [301] reported the anticancer effect of DOX-conjugated carboxymethylcellulose sodium (CMCNa) coated-γ-Fe_2_O_3_ NPs, through the inhibition of tumor cell proliferation, cell membrane disruption and induction of human breast cancer cells’ death. In addition, Plichta et al. [302] observed a 10–20% decrease in the survival rate of human cervix carcinoma cells (HeLa cell line) and human osteosarcoma cells (MG-63 cell line) under the action of DOX-conjugated polymer-coated γ-Fe_2_O_3_ NPs compared to free DOX treatment. Quan et al. [303] developed human serum albumin (HSA)-coated Fe_2_O_3_ NPs (HINP) conjugated with DOX and observed that in a 4T1 murine breast cancer xenograft model, DOX-HINP induced a reduction in tumor growth comparable to Doxil (a liposome-based DOX formula used as a treatment for various types of cancer) and superior to free Dox. This increased antitumor effect of magnetic NPs coupled with DOX is probably due to the activation of the hydroxyl radicals, which in turn damages mitochondria, lipids, proteins, DNA and other structures found in the cancer cells, leading in the end to their apoptosis and necrosis [10].

#### 3.2.5. Antitumor Effects of MgO NPs

Another class of biomaterials with a strong antitumor effect consists of MgO-based NPs. Mubarakali et al. [304] investigated their effect on human breast cancer MCF-7 cells and the results indicated an inhibition of the cell proliferation rates accompanied by specific cytomorphological characteristics of apoptosis. Moreover, Karthik et al. [305] evaluated the cytotoxic activity of MgO NPs against the A549 cancer cell line and it was observed that, by increasing the NPs’ concentration, the percentage of dead cells gradually grew up to almost 50%. Similarly, in another study, MgO NPs showed a strong toxic effect against A549 lung carcinoma cells through the increase in ROS, which in turn damaged the mitochondrial membrane potential and activated the apoptotic pathways [188]. In addition, due to the chemical stability of the MgO NPs (as obtained or with further modifications) under harsh conditions, their high tolerability in the human body and biodegradability [306,307,308,309,310,311], these biomaterials can be used with success in drug-delivery applications.

#### 3.2.6. Antitumor Effects of CuO NPs

Data found in the literature also reported on the anticancer effect of CuO NPs [7]. For example, these NPs showed a cytotoxic effect on human lung cancer cells and breast cancer cells, through the induction of apoptosis via enhanced production of ROS [312]. In another study, CuO NPs were used to treat mouse subcutaneous melanoma and metastatic lung tumors, based on B16-F10 mouse melanoma cells, through intratumoral and systemic injections, respectively [313]. The observations suggested that this type of NPs was capable of downsizing the growth of melanoma, inhibiting the metastasis of B16-F10 cells and increasing the survival chances of the mice models. Furthermore, the in vitro results on HeLa cells indicated that CuO NPs affected the mitochondria, which resulted in the release of cytochrome C and the activation of caspase-3 and -9, therefore inducing cell death. Moreover, CuO NPs were reported to possess a cytotoxic effect on human liver carcinoma cells in a dose-dependent manner, via ROS overproduction and, subsequently, induced oxidative stress [314].

#### 3.2.7. Antitumor Potential of TiO_2_ NP

TiO_2_ NPs are a prevalent material used in various biomedical applications, including cancer treatment [10]. Photocatalyzed TiO_2_ NPs have been reported as a potential strategy for cancer cell eradication. In one in vivo study, TiO_2_ NPs exposed to light irradiation suppressed tumor growth in glioma-bearing mice and increased the survival rate of the mice models [315]. Furthermore, nitrogen-doped anatase NPs demonstrated a higher visible light absorbance in comparison to the neat TiO_2_ NPs, inducing an almost 93% cell death of melanoma cells under UV light [316]. Similar results were observed in another study with colloidal ruthenium complex-loaded TiO_2_ NPs against melanoma cancer cells [317] and the results indicated that, under UV light, the number of dead cells increased in comparison with visible light illumination. However, the in situ penetration of UV light is low and dangerous to the human organism, therefore, a strategy to overcome this limitation is represented by the surface-functionalization of the TiO_2_ NPs. Recently, the efficacy of NIR on crystallized shells comprised of TiO_2_ NPs coated in order to form core/shell nanocomposites has been reported against HeLa cells and in a tumor model using female Balb/c nude mice [318]. Moreover, the study of Lucky et al. [319] reported the use of core-shell up-conversion nanoparticles with a thin and continuous layer of TiO_2_ against oral squamous cell carcinoma and their ability to reduce the in vivo generation of tumors. Venkatasubbu et al. [320] developed PTX-loaded HA/TiO_2_ NPs and evaluated their antitumor activity in diethylnitrosamine (DEN)-induced hepatocarcinoma in animal models and observed an enhanced anticancer activity for the modified PTX-loaded HA/TiO_2_ NPs compared to pure PTX. Similarly, in another study, the treatment of ovarian cancer cells with hyaluronic acid-TiO_2_ NPs loaded with cisplatin resulted in an improvement of intracellular drug accumulation when compared to free cisplatin [321]. Moreover, through the adjustment of the nanocarriers’ size and shape, a possible increase in drug accumulation in the microenvironment could be achieved, and various in vivo studies have demonstrated that elongated nanocarriers can be retained more efficiently at the tumor sites after intravenous injection and can deliver larger quantities of therapeutic drugs [322,323,324]. In this context, non-spherically shaped TiO_2_ nanoparticles with an elongated geometry could represent a possible drug-delivery strategy with higher efficiency. For example, Kafshgari et al. [325] fabricated well-separated, uniformly shaped and easily detachable anodic TiO_2_ nanotubes (NTs) and nanocylinders (NCs) through a time-varying electrochemical anodization protocol to investigate their potential application in cancer therapy. Accordingly, the newly fabricated nanotubes and nanocylinders were conjugated with DOX and their cellular uptake and cytotoxicity in HeLa cells were evaluated. The reported data indicated that the single and uniformly shaped pH-responsive anodic TiO_2_ NTs and TiO_2_ NCs possessed low cytotoxicity. When conjugated with the antitumor agent, they were easily incorporated into the cells and subsequently released their drug cargo into acidic intracellular compartments. In another study, Fe_2_O_3_ NPs-loaded TiO_2_ NTs were designed with the purpose of magnetic targeted guidance and site-specific drug delivery and the results suggested that the nanocarriers could be controlled and guided toward the cancer cells through a static gradient magnetic field [326]. In addition, the site-specific delivery of incorporated drugs was demonstrated through the nanocarriers’ conjugation with camptothecin, when a 90% killing efficiency of HeLa cells was achieved and with oligonucleotides for cell transfections demonstrating a 100% cellular uptake [326].

#### 3.2.8. Antitumor Effects of Other Metal Oxide Nanoparticles

The antitumor activity of NiO NPs has been lightly recorded in the specialized literature. Abbasi et al. [327] evaluated the anticancer effects of NiO NPs against the Hep G2 cancer cell line and observed that the pathogenic cells treated with increasing concentrations of nanoparticles exhibited a decrease in their survival rate that was dose-dependent. Thus, the highest concentration of NiO NPs induced a reduction in the survival rate by up to 84%, results that indicated a strong anticancer potential for this type of NPs. Similarly, Lingaraju et al. [130] synthetized via a green route NiO NPs and investigated their antitumor potential against A549 and Hep G2 cell lines. The results showed a dose-dependent cytotoxic effect against the cells treated with different concentrations of NiO NPs, probably due to the internal accumulation of nanoparticles and high stress, which in the end led to cellular death via apoptosis. Moreover, Zhang et al. [328] used green-synthesized NiO NPs to evaluate their anticancer activity on various tumor cell lines and the reported data indicated that the newly obtained NPs could decrease the viability of esophageal cancer cells up to 50% compared to other cancer cell lines such as colon cancer cells.

As with NiO NPs, the number or relevant works regarding the antitumor activity of ZrO NPs is very limited, despite their biomedical potential. One study investigated the cytotoxic effect of newly synthesized ZrO NPs against the MCF-7 cell line and the reported data suggested that, compared to the control group, the survival rate of the MCF-7 cancer cells was reduced in a dose-dependent manner [131]. Tabassum et al. [329] evaluated the cytotoxicity of ZrO NPs against the MDA-MB-231 cell line and observed a diminishing trend in the cells’ survival rate that was directly linked to the increase in the nanoparticles’ concentrations. In another study, a dose-dependent reduction in the survival rates of MDA-MB-231 and Hep G2 cancer cells was observed after exposure for 72 h to iron-manganese-doped sulfated ZrO NPs [330]. Another cell line with a high sensitivity to ZrO NPs is the A549 cell line, where in a study by Balaji et al. [331] after 24 h of treatment, a reduced level of viable cells could be observed.

As stated above, CdO NPs possess the ability to control cancer cells via the destruction of their cellular membrane, but this antitumor potential is rarely studied and reported in the specialized literature, with only a few articles investigating the cytotoxic effect against cancer cells. One such study, by Skheel et al. [32] evaluated the cytotoxic effects of green-synthetized CdO NPs against human colon cancer cells (HT29) and the obtained data demonstrated their significant antitumor effect against the studied cancer cells. Moreover, Gowri et al. [33] showed that, even at a minimal concentration, ZrO NPs are able to induce an inhibitory effect against human cervical cancer cells, thus implying their promising potential as anticancer agents.

### 3.3. Antibacterial Activity

The ever-increasing resistance of different pathogens towards antibiotics coupled with the need for biomedical devices, such as implants, wound dressings, catheters and stents, with a wide range of antibacterial activity, forced researchers to identify and develop new strategies in the battle against various bacterial agents. In the last 20 years, nanotechnology has offered a solution in the form of a variety of nanoparticles that have been proven by in vitro and in vivo studies to possess antibacterial effects (Table 3). Of these, metal oxide nanoparticles such as Ag_2_O, TiO_2_, CuO, ZnO and MgO have been identified to exhibit antibacterial activity against several bacterial species [332]. 

Literature data indicate that MONPs can interfere with different cellular processes of pathogens, either through the generation of ROS, which causes oxidative stress, or either through their dissolution and release of toxic free metal ions [333] (Figure 6). In addition, the antibacterial effectiveness of MONPs is dictated by their characteristics, such as size and surface properties, determined by their synthesis parameters [334,335].

**Table 3 jfb-13-00274-t003:** Metal oxide NPs with antibacterial activity: mechanism of action and characteristics.

MONPs	Mechanism of Action	Factors that Influence MONPs Effectiveness	Antibacterial Activity and Characteristics	Ref.
Ag_2_O	dissolution and ion release from the NPs surface → pits and gaps in the bacteria membrane → disruption of the metabolic processes due to ion interaction with disulfide or sulfhydryl enzymes → DNA damage	NPs shape and size	action on drug-resistant bacteria; high stability	[336,337,338,339,340]
CeO	dissolution and ion release → ROS generation → DNA damage → cellular death	NPs concentration and surface properties	efficient against both Gram-positive/Gram-negative bacteria	[341,342]
CuO	dissolution and ion release → vital enzyme damaging	NPs size and shape	efficient against Gram-positive/Gram-negative bacteria; high stability	[343,344,345]
MgO	loss of cell membrane integrity → extracellular leakage of intracellular contents → cellular death	NPs concentration, pH and size	efficient against both Gram-positive/Gram-negative bacteria; high stability; low cost	[346,347,348]
TiO_2_	ROS overproduction → oxidative stress → lipid peroxidation → membrane fluidity	NPs size, shape and crystal structure	high stability	[349,350,351,352]
ZnO	dissolution and ion release + ROS generation → membrane dysfunction → NPs internalization into the cell	NPs concentration and size	efficient against both Gram-positive/Gram-negative bacteria; high stability; effectiveness against spores	[353,354,355]

For example, it was demonstrated that smaller nanoparticles exhibited a stronger bactericidal effect compared to both bigger NPs and their bulk counterpart [333,336,356,357], while NPs with positively charged surfaces possessed a stronger binding force for the negatively charged surfaces of various bacterial agents, therefore leading to an enhanced antibacterial activity [334].

#### 3.3.1. Antibacterial Activity of ZnO NPs

The antibacterial activity of ZnO NPs stems from their high solubility and Zn^+^ ion release, which, once in contact with the bacterial cells, will end up being absorbed. At this intracellular level, the free Zn^+^ ions will interact with the thiol group of respiratory enzymes and inhibit their action, leading to an overproduction of ROS and free radicals, which in turn will cause oxidative stress. This will result in membrane, mitochondria and DNA damage and ultimately bacterial cell death [358]. Data from the specialized literature indicate the bactericidal effect of ZnO NPs against both Gram-positive and Gram-negative bacteria, but also against spores, which in general possess resistance against high pressures and temperatures [354]. Moreover, it was reported that the antibacterial activity of ZnO NPs is tightly related to their size and concentration. For example, (*i*) Padmavaty et al. [359] showed that the bactericidal activity of ZnO NPs increased with the decrease in particle size, while (*ii*) Zhang et al. [249] demonstrated that the bacterial response to ZnO NPs is both dose- and time-dependent, with positive results being observed in low-dose ranges and smaller exposure periods; similarly, (*iii*) Hosseinkhani et al. [360] reported a decrease in the number of bacteria with the decrease in the NPs size, while (*iv*) Emami-Karvani et al. [361] investigated the effect of ZnO NPs against both Gram-positive (*Escherichia coli*) and Gram-negative (*S. aureus*) bacteria and the results indicated that the antibacterial activity of ZnO NPs is both size- and concentration-dependent. Furthermore, several studies investigated the synergistic action of ZnO NPs with antibiotics as an alternative treatment for various bacterial diseases. In this context, Ghasemi and Jalal [362] evaluated the effect of ZnO NPs on the effectiveness of two antibiotics, namely ciprofloxacin and ceftazidime, against an opportunistic pathogen, *Acinetobacter baumannii*, which causes a wide range of diseases (e.g., meningitis and pneumonia) and reported that, in the presence of sub-inhibitory concentrations of ZnO NPs, the antibacterial activity of both antibiotics was enhanced. In addition, ZnO NPs have been demonstrated to possess antibacterial activity against *Vibro cholerae* [363], *Camphylobacter jejuni* [364], *Mycobacterium tuberculosis* [365], etc., namely to pathogenic agents responsible for a wide range of illnesses such as severe watery diarrhoea, dysentery and tuberculosis.

#### 3.3.2. Antibacterial Activity of TiO_2_ NPs

The antibacterial properties of TiO_2_ NPs are associated with their specific characteristics such as crystal structure, size and shape [350], with the proposed mechanism of action being correlated to their ability to generate ROS and cause DNA damage [351,366]. Thus, in a study conducted by Roy et al. [351], it was reported that TiO_2_ NPs were able to improve the antibacterial activity of a wide range of antibiotics, such as cephalosporins, tetracycline, glycopeptides and macrolides, against *methicillin-resistant S. aureus* (MRSA). In addition, it was demonstrated that the photocatalytic properties of TiO_2_ NPs facilitate the eradication of bacteria. For example, Carré et al. [352] suggested that the antibacterial photocatalytic activity of these NPs is accompanied by lipid peroxidation that causes membrane fluidity enhancement and lowers the cell’s integrity. However, despite the enhancement offered by ultra-violet (UV) light exposure, the use of TiO_2_ NPs under UV light is limited due to the genetic damage observed in human cells and tissues [349].

#### 3.3.3. Antibacterial Potential of CuO NPs

Due to their unique physicochemical and biological characteristics, antibacterial activities and low cost of preparation, CuO NPs have attracted the attention of researchers all over the world [Wu et al., 2002; Usman et al., 2013]. Mahapatra et al., 2008, investigated the antibacterial activity of CuO NPs against various bacteria including *Salmonella paratyphi*, *Shigella strains*, *Klebsiella pneumonaie* (*K. pneumonaie*) and *P. aeruginosa* and reported that these nanoparticles were able to reduce the number of bacteria through membrane crossing and enzyme damage, which in turn led to cell death. In another study, Azam et al., 2012, evaluated the effect of CuO NPs against two Gram-positive bacteria and two Gram-negative bacteria and the results showed that the antibacterial effect against both groups of bacteria was size- and concentration-dependent. Moreover, Ahamed et al. [345] showed that the CuO NPs exhibited significant antibacterial activity against a wide range of bacterial strains such as *Enterococcus faecalis*, *K. pneumonaie*, *E. coli*, *P. aeruginoasa*, *Shigella flexneri*, *S. aureus*, *Proteus vulgaris*, *S. typhimurium*, etc.

#### 3.3.4. Antibacterial Activity of MgO NPs

Data from the literature highlight a strong antibacterial activity of MgO NPs, which can be correlated to their ability to generate superoxide on their surface and increase the pH of the microenvironment by particle hydration with water [348]. In addition, Jin and He [346] demonstrated that this type of MONPs is capable of damaging the membrane, resulting in a leakage of the intracellular contents and, in the end, cell death. MgO NPs exhibit antibacterial effects against both Gram-positive and Gram-negative bacteria [347]. For example, Sawai et al. [367] reported that MgO NPs possess antibacterial effectiveness against *S. aureus* and *E. coli*, while another study proved the particles’ efficiency against *E. coli* and *Salmoella stanely*, but in a concentration-dependent manner [346]. Vidic et al. [347] investigated the antibacterial potential of a combined nanostructure of ZnO-MgO and their results indicated a high antibacterial activity against the Gram-positive bacterium, *B. subtilis*, while the pure MgO NPs revealed a moderate activity against both *B. subitlis* and *E. coli* bacteria.

#### 3.3.5. Antibacterial Potential of Ag_2_O NPs

Ag_2_O NPs have also been discovered to possess great antibacterial effectiveness against both regular and drug-resistant bacteria, turning them into potential novel alternatives to most in-use antibiotics [349]. In a study by Sondi and Salopek-Sondi et al. [368], it was suggested that the antibacterial mechanism of action for Ag_2_O NPs is due to their ability to induce cell death through oxidative stress caused by arresting the cell cycle in the G_2_/M phase as a direct consequence of DNA damage.

#### 3.3.6. Antibacterial Activity of CeO_2_ NPs

In terms of CeO_2_ NPs, there is widespread research on their antibacterial action [369,370,371] with studies showing their effectiveness against nitrogen-fixing bacteria and Gram-negative bacteria. In contrast to other MONPs, it was discovered that CeO_2_ NPs cannot penetrate the bacteria membrane [341]; therefore, the hypothesized mechanism of action is based on the generation of oxidative stress in lipids and proteins found in the plasma membrane and on the disruption of the electron-flow and bacterial respiration [155]. Ravishankar et al. [342] reported that CeO_2_ NPs exhibited a concentration-dependent antibacterial activity against *P. aeruginosa* and a lack of activity against Gram-positive bacteria.

#### 3.3.7. Antibacterial Activity of Other Metal Oxide Nanoparticles

Even though the exact mechanism of the NiO NPs antibacterial activity is not yet fully understood, data reported in the literature indicate their antibacterial effect against various Gram-positive and Gram-negative bacteria such as *B. subtilis*, *S. aureus*, *E. coli* and *P. aeruginosa* [30]. For example, in one study, the newly developed NiO NPs were shown to have strong antibacterial activity against *S. aureus* and *B. subtilis*, while the least-susceptible strains were found to be *P. aeruginosa* and *K. peneumonaie* [327]. Moreover, Lingaraju et al. [130] demonstrated the nanoparticles’ strong antibacterial activity against *E. coli* and only a moderate effect on the *S. aureus* and *K. aerogenes* strains.

As with other MONPs, ZrO_2_ NPs exhibit an antibacterial effect on both Gram-positive and Gram-negative bacteria. Gowri et al. [372] evaluated the antibacterial potential of the ZrO_2_ NPs against *E. coli* and *S. aureus* and observed that the inhibition zones in the case of cotton fabrics treated with ZrO_2_ NPs were greatly improved. Furthermore, Kumaresan et al. [373] revealed the nanoparticles’ ability to inhibit the growth of bacteria strains such as *E. coli, Salmonella typhi* and *B. subtilis*. In another study, *Penicillium* was used to synthesize ZrO_2_ NPs and it was observed that, even at minimum concentrations, the nanoparticles had strong antibacterial activity against *P. aeruginosa* and *E. coli* [374].

The antibacterial activity of CdO NPs was tested against three Gram-positive (*S. aureus, S. pneumoniae* and *B. subtilis*) and three Gram-negative bacteria (*Proteus vulgaris*, *P. aeruginosa* and *S. typhi*) at three different concentrations. The results indicated that the maximum zone of inhibition was obtained against *P. vulgaris* suggesting that this bacterial strain is more susceptible to CdO NPs, and this can be explained through structural differences in the cell membranes. Moreover, *S. aureus* showed the least zone of inhibition, thus indicating the reduced antibacterial activity of these NPs on this bacterial strain [33]. In addition, Shkeel et al. [375] prepared CdO NPs with the use of aqueous plant extracts from the Curcuma rhizome and investigated their antibacterial effect against a variety of human pathogens such as *P. aeruginosa, K. pneumoniae, S. aureus*, *E. coli, Candida albicans* and *Trichophyton rubrum* and reported a superior activity in the area of antimicrobial strain inhibition.

Considering the strong antibacterial activity of MONPs, their use in combination therapy could represent a feasible strategy to overcome the current rise in bacterial resistance and biomedical device-mediated infection. However, further studies are still required in order to minimize the toxicity of nanoparticles before they can be employed as potential alternatives for antibiotics and disinfectants for biomedical applications.

## 4. Toxicological Effects of Metal Oxide Nanoparticles

Due to their unique and novel characteristics, MONPs have become the main focus of a wide range of in vitro and in vivo studies, consequently turning them into potential but powerful scientific tools, with diverse applicability in the biological field [241]. However, despite their multiple therapeutic effects, the potential toxicity of MONPs has been a rising concern, as numerous in vitro and in vivo studies report controversial results [376]. Different toxicity mechanisms, such as oxidative stress/lipid peroxidation/cell wall damage as a result of an overproduction of ROS [377,378], metal ion release [379,380,381] and an impaired interaction between the nanoparticles and the targeted cells [382,383,384], have been suggested (Figure 7). However, no singular, indisputable and precise identification of the toxicity mechanism involving MONPs has been made.

Furthermore, it is a well-known fact that the toxicological risk of any given substance is determined both by its inherent toxicity and exposure time. Therefore, if the exposure time is kept to a minimal threshold, the toxicity risk might be reduced even though the MONPs possess a powerful cytotoxic effect [385]. Owing to the ever-growing importance of MONPs in the fields of biology and medicine, an incremental need for risk assessment of the toxic effect of MONPs should be further studied and characterized. The following section will outline the most recent in vitro and in vivo toxicity studies of various MONPs synthetized via different techniques.

### 4.1. In Vitro Studies

Despite being recognized as one of the most widely used non-toxic mineral nanostructures, TiO_2_ NPs exhibit various specific properties that could possess a potentially unknown cytotoxic effect toward the human organism. Several in vitro studies reported the toxic effects of TiO_2_ NPs in various cell types such as COS-1, NIH-3T3, TK6, WIL2-NS, NIH-3T3, etc. [386,387,388,389]. Magdolenova et al. [389] reported that the cytotoxic effects displayed by the TiO_2_ NPs are mainly attributed to the type of dispersion procedure used during toxicological investigations, while Kang et al. [387] and Akhal’tseva et al. [390] observed DNA damage and micronuclei generation in human lymphocytes and the human lymphoblast TK6 cell line after TiO_2_ NPs treatment. However, since different testing protocols influence a variety of nanoparticle interactions, it may have a divergent impact on the toxicity results and the data reported in the literature are inconsistent and often conflicting with numerous in vitro studies disconfirming the cytotoxic potential of TiO_2_ NPs [391]. In common with TiO_2_ NPs, ZnO NPs also present a widespread range of applicability in biomedicine, with numerous studies reporting on their multiple therapeutic benefits [11]. However, in the last few years, the use of ZnO NPs has become debatable, mainly after it was discovered that they induce an over-accumulation of ROS and subsequently cause cytotoxic effects on certain specific organs and cell lines [11]. Data reported in literature assign the biotoxicity of these nanoparticles on their high solubility, a property responsible for the increase in the free intracellular Zn^2+^ ion concentration released into the microenvironment [11]. As a direct consequence, in an in vitro system, the cellular Zn homeostasis disruption has been associated with oxidative stress, mitochondrial dysfunction and, ultimately, higher toxicity [392] (Figure 6). Nonetheless, important aspects that should be taken into consideration when assessing ZnO NPs toxicity are the concentration, the time of exposure to the particles and whether or not their toxicity could be reversible [258]. For example, Sahu et al. [393] reported that exposure to 50 nm ZnO NPs at concentrations of between 5 and 100 μg/mL decreased cell viability through the generation of oxidative stress, in a concentration-dependent manner, ultimately resulting in DNA damage and cell apoptosis. Moreover, Huang et al. [394] showed that 20 nm ZnO NPs exerted a concentration- and time-dependent cytotoxic effect on BEAS-2B human lung epithelial cells; while Wu et al. [395] and Ng et al. [396] demonstrated that the nanoparticles’ cytotoxic effect on the primary human bronchial epithelial cell line BEAS-2B was induced by 24–70 nm ZnO NPs and 22.5 nm ZnO NPs, respectively.

In addition to specific cell lines derived from the airways, a vast number of in vitro studies focused on the toxic effects of ZnO NPs on various immune cells, such as macrophages and monocytes. In the RAW 264.7 macrophage-like cell line, 20 nm ZnO NPs led to the induction of intracellular Ca^2+^ flux, a process followed closely by reduced mitochondrial membrane potential and, finally, to the loss of membrane integrity [397]. However, similarly sized ZnO NPs did not activate the inflammasome in the THP-1 human monocytic cell line, therefore, no cytotoxic effects could be observed. Moreover, it was reported that monocytes are more sensitive to 4–20 nm sized ZnO NPs and that with NPs size reduction, the cytotoxicity and ROS generation were significantly increased [398]. As stated above, the manifold effects of CeO_2_ NPs have motivated researchers all over the world to pursue these MONPs as therapeutic agents for a number of diseases, including cancer, with various in vitro and in vivo studies demonstrating their cytotoxic effects on a wide range of tumor cells [281]. However, data published in the specialized literature indicate that CeO_2_ NPs can also display minimal in vitro toxicity to non-cancerous cells, most likely due to the impact of various unknown cellular and micro-environmental stimuli on the manifestation of anti- and pro-oxidant behavior [281]. For example, several in vitro reports showed that the uptake of CeO_2_ NPs could induce oxidative stress, DNA damage, aberrant cell signaling, substrate dephosphorylation and modifications at the transcriptional and post-translational levels [277,399,400]. However, due to the distinct properties of the synthetized ceria NPs and the poor correlation between their cytotoxic effects with particle size/surface characteristics and cell type, the majority of the reported data regarding their toxicity are often contradictory [376]. As with other MONPs, the intracellular and in vivo toxicity of Fe_2_O_3_ nanoparticles arise from the over-production of ROS, which can damage cells by peroxidizing lipids, disrupting DNA, modulating gene transcription, changing proteins and ultimately lowering the physiological function of cells, followed by their death (Figure 6). The cytotoxic effect of Fe_2_O_3_ NPs has been investigated in vitro on a wide range of cell lines such as human lung epithelial cells, human epidermal keratinocytes, BRL3A rat liver cells, Cos-7 monkey fibroblasts, etc. [401], but reported end results were contradictory mainly due to the wide variety of cell lines employed. For example, amine-modified Fe_2_O_3_ NPs at a concentration of 224 µg/mL were shown to induce an up to 25% reduction in the astrocytes derived from a mouse brain, while human dermal fibroblasts and fibrosarcoma cells did not present the same drastic changes in their survival rate [402]. Moreover, the toxicity of Fe_2_O_3_ NPs has been demonstrated to be strongly dose-dependent, with concentrations higher than 300 µg/mL usually inducing cytotoxic effects after a prolonged exposure time [403,404]; Dwivedi et al. [358] reported that cell death is a dose-dependent phenomenon and it was associated with increasing concentrations of NPs, which led to the generation of ROS-mediated oxidative stress. In vitro studies are a step toward clinical practices, especially due to the reproducibility of material behavior and cellular response. However, one of their considerable challenges lies in the inability to replicate cellular conditions in ’live’ systems, which may curtail the data of in vitro test to effectively ascertain in vivo response, which will be looked into in the following section.

### 4.2. In Vivo Studies

Regardless of the in vitro toxicity evaluation of MONPs, in vivo studies are scarcely reported. This section presents the most relevant in vivo animal investigations concerning the toxic effects of these metal oxide nanoparticles. The in vivo toxicity of ceria NPs was evaluated in a series of studies that demonstrated that animal exposure to certain concentrations of CeO_2_ NPs led to lung inflammation, lung injury, alveolar macrophage functional alteration, induction of phospholipidosis and release of pro-inflammatory and fibrotic mediators [405]. Moreover, reported data suggest that ceria NPs can induce myocardial fibroblast proliferation and collagen accumulation in rat models [406]. In addition, CeO_2_ NPs could lead to systemic toxicity, since cerium is not a mineral found normally in the human body, therefore lacking an intrinsic clearance mechanism for its subsequent elimination [376]. As with CeO_2_ NPs, in vivo toxicity studies of TiO_2_ NPs showed stronger inflammatory activity in comparison to their micro-sized counterparts, leading to lung inflammation and, consequently, cancer in rats after nanoparticle inhalation and intratracheal instillation [388,407,408]. Moreover, Trouiller et al. [409] and Sycheva et al. [410] reported that TiO_2_ NPs administered either through drinking water or gavage could induce micronuclei and DNA damage in the peripheral blood cells and bone marrow of adult male mice, while Lindberg et al. [411] observed that, after five days inhalation of TiO_2_ NPs, no DNA damage occurred in the peripheral blood lymphocytes of mice. As stated above, the system distribution of ZnO NPs can lead to toxic manifestations in various organs of the body, based on their concentration, the administered dose and its route, and the exposure time. Taking this into consideration, a wide range of in vivo studies demonstrated that ZnO NPs could affect organs such as the liver, kidneys, spleen, stomach, pancreas, lungs and heart, but also the neurological and lymphatic systems. For example, ZnO NPs inhalation caused a size-dependent severe inflammatory response and fibrosis in the alveolar and tracheobronchial tissues, due to the dissolution of nanoparticles by the acidic lung fluid, which in turn increased their concentration and, implicitly, their pulmonary cytotoxicity [412]. Moreover, Han et al. [413] reported that the administration of ZnO NPs via intraperitoneal injection resulted in neurotoxic effects, such as an attenuated learning ability and memory. Similarly, Elshama et al. [414] identified significant histopathological and ultrastructural alterations in the brains and spinal cords of rats as a direct consequence of increased ROS after prolonged exposure to high doses of intraperitoneally administered ZnO nanoparticles. Furthermore, in a study by Li et al. [415], it was demonstrated that both intraperitoneal and oral administration of ZnO NPs had, as a direct consequence, the systemic distribution and toxic accumulation of nanoparticles in different organs such as the liver, lungs, kidneys and spleen. In addition, it was shown that in male rat models, the nanoparticles were able to alter the body’s metabolism by elevating the levels of liver enzymes. Similar results were reported by Soheili et al. [416], who, apart from the elevated levels of liver enzymes, observed higher levels of glucose, thus suggesting that ZnO NPs are also harmful to some extent to pancreatic cells. In another study, Esmaeilloua et al. [417] demonstrated that the cytotoxic effects of ZnO NPs on organs such as the lungs, liver and kidneys could be also dependent on their physicochemical characteristics, such as size and specific surface area. Similarly, Kim et al. [418] reported that the oral administration of a wide range of doses of 100 nm sized ZnO NPs with various surface charges and for a prolonged period of time were capable of inducing an assortment of histopathological alterations such as squamous and glandular cell hyperplasia in the stomach, acinar pancreatic cells apoptosis, retinal atrophy and suppurative inflammation in the prostate. Despite the scarcely available data on the effects of the ZnO NPs on the reproductive system and fetal development, Hong et al. [419] showed that the repeated oral administration of ZnO NPs for short periods of time in pregnant rat models led to maternal and developmental toxicity. When it comes to the in vivo interaction between Fe_2_O_3_ NPs and biological systems, the process becomes quite complicated and dynamic [420,421,422]. When these NPs are administered and enter an organism through various routes, their absorption can occur through interactions with different biological components, e.g., proteins and cells, only to be afterward distributed into different organs where they either remain or end up being metabolized [423]. Taking this into consideration, it was expected that organs that are enriched with reticuloendothelial systems, such as the lungs, liver and spleen, would take up the majority of iron oxide nanoparticles administered via different routes. In vivo studies demonstrated that Fe_2_O_3_ NPs that entered the body both via inhalation and the intravenous route, accumulated in the lungs, brain, spleen, testes and liver, while only intravenous Fe_2_O_3_ NPs were found to accumulate in the brain, spleen and kidneys [424,425]. In addition, different types of animal models also displayed different toxicity profiles for Fe_2_O_3_ NPs, e.g., differences in liver function in terms of the levels of aspartate aminotransferase and alanine aminotransferase, were observed in different types of animal models upon intravenous administration of the same concentration of iron oxide NPs [426].

In summary, the toxicity of MONPs remains an overly controversial subject, and further standardized studies are still required in order to understand the exact mechanism through which nanoparticles exert their cytotoxic effects and the ways through which this present liability could be overcome.

## 5. Conclusions and Future Perspectives

Due to their small size and unique characteristics, NPs possess different properties compared to their bulk counterparts, thus exhibiting an ever-growing biomedical potential, mainly due to their inherent ability to induce an over-production of ROS and implicitly cell death. These unique properties turn MONPs into suitable anticancer, antibacterial and pro-regenerative tools, either as standalone agents or as drug-delivery platforms for various bioactive molecules. However, despite the multiple advantages that MONPs bring into clinical practice, the major drawback that poses a great challenge for researchers all over the world is their inherent toxicity and the lack of a consensus regarding the guidelines and regulatory frameworks for in vitro and in vivo testing of their toxicological effects.

With this in mind, systematic documentation of the complementary in vitro and in vivo protocols used to evaluate specific NPs responses would be beneficial and more than likely eliminate the limitations imposed by their toxicity, especially if it would present both as a standardized method of synthesis and as testing protocols. From the point of view of synthesis and functionalization, firstly, due to the variety of synthesis methods, there is difficulty in evaluating the influence of the NPs size and morphology, and, secondly, as a result of differences in the chemical and surface properties of the same metal oxide NPs synthesized by different methods, there is a disparity in the performance and actual functionalization of such NPs.

Each MONP synthesis method comes with its own advantages, and from the different bottom-up approaches discussed in the present review, typically precipitation, sol-gel, hydrothermal and various biosynthesis methods are extensively used in both laboratory and industrial synthesis. Furthermore, biosynthesis or green synthesis possesses a huge potential for the advancement of green research in the field of MONPs with biomedical applications due to its effectiveness, low costs, sustainability and environmental and human health safety benefits.

Once this challenge of systematic documentation is addressed and overcome, the future of MONPs in the biomedical field will hold an even greater promise, not only for multifunctional therapeutic strategies but also for early disease diagnosis as theranostic platforms for in-depth and non-invasive cellular and tissue imaging. Moreover, recent advances in the design and development of MONPs allow them to deliver, with minimal effects, active drugs for lethal diseases that demand site-specific treatment. This characteristic was and will continue to be heavily exploited in cancer treatment, with researchers seeking multi-functional therapies and remote control of NPs functions. Furthermore, this will open the door for the introduction of personalized medicine, which will pave the way for safer, more effective and tailored treatment options for patients with cancer. Therefore, it is expected that the coming years should witness an increase in the number of clinical trials and an improvement in the life of cancer patients.

## Figures and Tables

**Figure 1 jfb-13-00274-f001:**
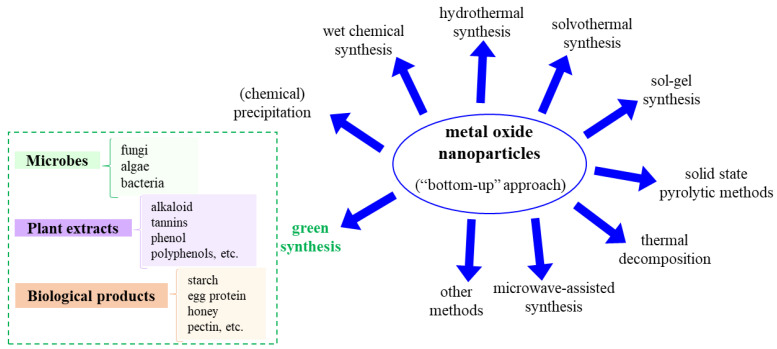
Possible metal oxide nanoparticle synthesis methods (“bottom-up” approach).

**Figure 2 jfb-13-00274-f002:**
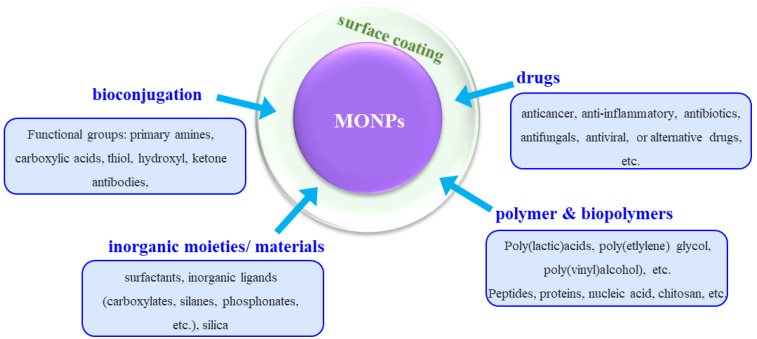
Schematic overview of the different surface modification/functionalization of MONPs usually applied for improving the biological effects.

**Figure 3 jfb-13-00274-f003:**
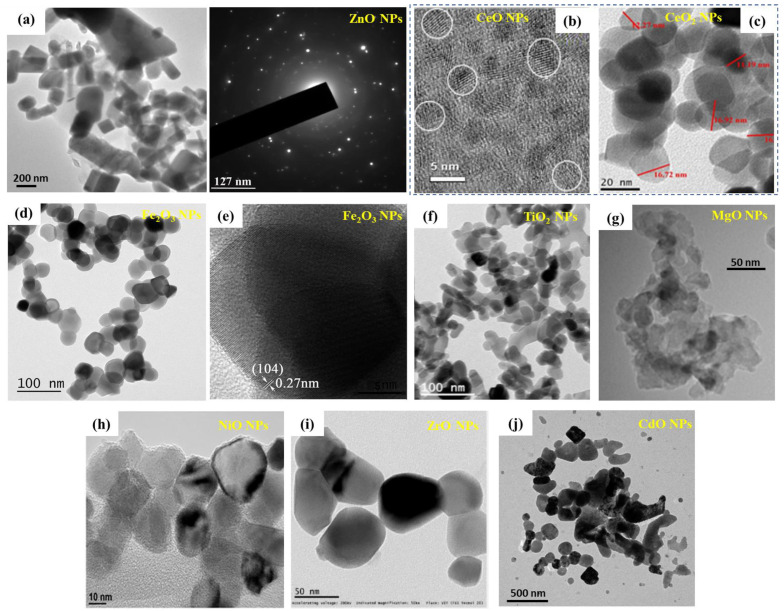
Morphology of MONPS: (**a**) HRTEM image showing green-synthesized ZnO NPs (*E. prostrata* leaf extract) together with the SAED pattern (reprinted from ref. [177]. (**b**) HRTEM image demonstrating spherical crystalline CeO NPs, diameter ~5 nm, representative crystallites with lattice fringes in white circles (reprinted with permission from ref. [178]. Copyright 2021 Elsevier). (**c**) HRTEM image of green-synthesized 20 nm CeO_2_ NPs (hydrothermal method mediated by *E. globulus* leaf extract (reprinted from ref. [180]). (**d**,**e**) green-synthesized α-Fe_2_O_3_ nanoparticles (guava leaves, Psidium guajava): (**d**) TEM micrograph, (**e**) HRTEM image of a single nanocrystal showing lattice fringes with a spacing of 0.27 nm (reprinted with permission from ref. [181]. Copyright 2016 Royal Society of Chemistry). (**f**) TEM image of TiO_2_ NPs obtained by thermal decomposition (reprinted from ref. [94]). (**g**) HRTEM image of MgO NPs obtained by a sol-gel method (reprinted from ref. [70]. Copyright© 2019 Alfaro et al.). (**h**) TEM image of hydrothermal NiO NPs (reprinted from ref. [182]. Copyright 2017 AIP Publishing). (**i**) HRTEM image of ZrO NPs obtained by a green synthesis (reprinted from ref. [131]. Copyright 2017 Elsevier). (**j**) TEM image of CdO NPs obtained by the annealing of formed complexes (reprinted from ref. [183]).

**Figure 5 jfb-13-00274-f005:**
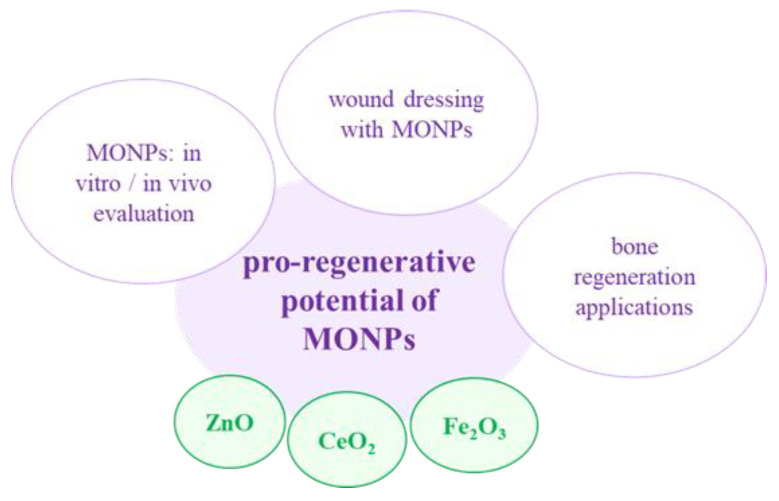
MONPs and their use in pro-regenerative potential.

**Figure 6 jfb-13-00274-f006:**
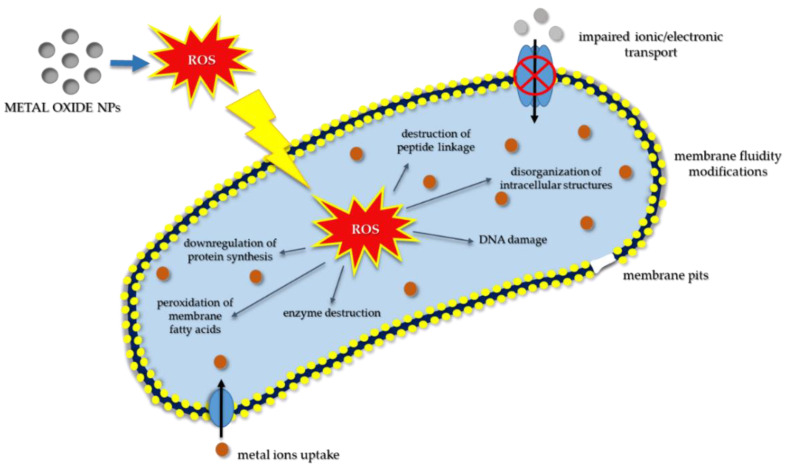
Schematic representation of various mechanisms of antibacterial activity of the MONPs, illustrating the possible interactions of MONPs with the bacteria cell wall, membrane structures, DNA, enzymes and other proteins and the influence of ROS on membrane stability, protein function and synthesis.

**Figure 7 jfb-13-00274-f007:**
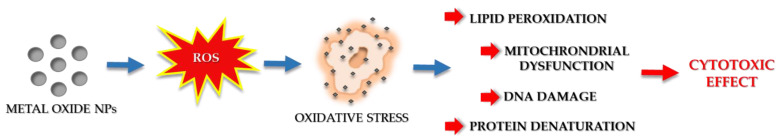
Schematic representation of the possible toxicity mechanisms of metal oxide nanoparticles.

**Table 1 jfb-13-00274-t001:** Supplementary in vitro and in vivo studies focusing on evaluating the anticancer activity of ZnO NPs as standalone agents or as drug carriers.

MONPs	Cell Line	Observations	Ref.
ZnO NPs	human prostate cancer cells (PC3 cell line); non-small cell lung cancer cells (A549 cell line)	↓ serum levels of tumor markers; ↓ levels of hepatocyte integrity and oxidative stress markers	[256]
	human skin melanoma (B16F10 cell line); human skin melanoma (A375 cell line)	↓ ERK (extracellular signal-regulated kinases) enzyme and other cancer-associated kinases	[257]
	human colon carcinoma (LoVo cell line)	severe oxidative stress → DNA damage	[258]
	human cervical carcinoma cells (HeLa cell line)	↑ mRNA expression of p53; ↑ levels of ROS	[241]
	hepatocellular carcinoma (Hep-G2 cell line)	dose-dependent cytopathic effects	[177]
	Caco-2	↑ ROS and 8-oxodG levels; micronuclei and DNA damage	[259]
	cervical cancer cells (SiHa cell line)	dose-dependent cytotoxic effects via mitochondria apoptotic and necrotic death	[260]
	human breast cancer cells (MCF-7 cell line)	dose-dependent cytotoxicity against tumor cells when treated for 24 h	[261]
	human breast cancer cells (MDA-MB-231 cell line)	dose-dependent antitumor activity within the concentration interval of 12.5–200 µg/mL after a 24 h treatment	[246]
	Human skin melanoma (A375 cell line)	↓ cell viability; ↑ROS generation; ↑ apoptosis confirmed by chromosomal condensation assay and caspase-3 activation; ↓ density and a round morphology	[262]
	human lung cancer cells (A549 cell line)	dose-dependent cytotoxic effect via ROS overproduction; ↑ membrane damage; ↑ oxidative stress; ↓ mitochondrial membrane potential	[263]
	wild type EGFR A549 and EGFR-mutated CL 1–5 cells	↑ cytotoxic effect in CL 1–5 cells compared to A549 cells	[264]
ZnO/SiO_2_ core-shells NPs	human prostate adenocarcinoma (LNCaP and Du145 cell lines)	↑ radiation-induced reduction in the cell survival rate	[265]
PEG-ZnO NPs	breast cancer cells (MCF-7; MDA-MB-231; MDA-MB-468; T-47D cell lines)	↑ cytotoxic effect against all breast cancer cells at a concentration of 25 µg/mL when treated for 24 h	[244]
daunorubicin-ZnO NPs	human lung cancer cells (A549 cell line)	no premature drug leakage; relevant therapeutic concentrations; liposome incorporated NPs demonstrated a pH-responsive release of the active drug	[266]
	sensitive leukaemia cells (K562 cell line) resistant leukaemia cells (K561/A02)	↑ sensitivity of the drug-resistant tumor cell line; ↑ accumulation of daunorubicin; ↑ cell membrane permeation; ↑ uptake of daunorubicin into both cell types	[267]
DOX-ZnO NPs	human breast cancer cells (MCF-7 cell line)	↑ cytotoxicity for the drug-loaded NPs	[53]
isoorientin-DOX ZnO NPs	hepatocellular carcinoma (HepG-2)	dose- and time-dependent antitumor activity; ↑ synergistic anticancer activity; ↑ cell death through mitochondrial dysfunction; Akt and ERK1/2 inhibited phosphorylation and JNK and P38 enhanced phosphorylation; no significant damage to normal healthy liver cells	[268]
photofrin-ZnO NPs	human lung cancer cells (A549 cell line)	↑ ROS production and cell death	[269]
Folic acid (FA)-functionalised-PTX-ZnO NPs	human breast cancer cells (MCF-7 and MDA-MB-231)	combined passive and active targeting of paclitaxel; ↑ efficacy of paclitaxel against subcutaneous tumors in vivo	[270]
DOX-ZnO/PEG nanocomposites	human cervical cancer cells (HeLa)	↑ antitumor activity; ↑ cancer cell injury via ROS under UV irradiation; ↑ intracellular concentration of DOX with an enhanced anticancer effect	[271]

↑ indicates enhancement; ↓ indicates inhibition.

**Table 2 jfb-13-00274-t002:** Overview of in vitro and in vivo studies focusing on evaluating the anticancer activity of CeO_2_ NPs as standalone agents or as drug carriers.

MONPs	Cell Line	Observations	Ref.
CeO NPs	human lung cancer cells (A549 cell line)	↑ free-radical production; ↑oxidative stress; ↑ cytotoxic effect against cancerous cells	[282]
	human prostate cancer cells (PC-3 cell line)	↑ antitumor affect against cancer cells; no injury caused to normal healthy cells	[284]
	ovarian cancer cells (A2780 cell line); A2780 xenograft murine model	concentrations in the range of 25–50 µM exhibited an anti-angiogenic effect in ovarian cancer cells; ↓ tumor size in vivo	[280]
	human colon cancer cells (HCT 15 cell line)	↓ cell viability via ROS overproduction	[283]
	human neuroblastoma cells (IMR31 cell line)	↑ ROS production that resulted in oxidative stress; ↑ cytotoxicity and genotoxicity	[285]
	Fibrosarcoma (WEHI146 cell line)	↑ antitumor activity by increasing ROS generation and inducing apoptosis	[286]
camptothecin-CeO_2_ NPs	human pancreatic cancer cells (BxPC-3 cell line)	↓ cell viability	[287]
chlorin e6-CeO_2_ NPs	Human breast cancer cells (MCF-7/ADR cell line); MCF-7/ADR xenograft murine model	exhibited photodynamic therapy against drug-resistant breast cancer cells and in vivo tumors	[44]
DOX-CeO_2_ NPs	human ovarian cancer cells (A2780; SKOV-3 and CAOV-3 cell lines)	↓ cell proliferation rates; ↑ apoptosis of cancerous cells compared to free DOX	[288]
curcumin-CeO_2_ NPs	neuroblastoma cells (IMR-32; SMS-KAN; SK-N-AS, LA-N-6)	induced significant cell death in all of the mentioned cancer cells	[289]
DOX-CeO_2_ NPs	human liver cancer cells (HEPG-2 cell line)	induced a synergistic antitumor activity on cancer cells	[290]

↑ indicates enhancement; ↓ indicates inhibition.

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
