# Peer review of "Metal Oxide Nanoparticles: Review of Synthesis, Characterization and Biological Effects"

_jfb, 2022, doi:10.3390/jfb13040274_

Round 1

Reviewer 1 Report

The manuscript presented for evaluation concerns methods for obtaining metal oxide nanoparticles and their biomedical applications. In my opinion, the work should be rewritten. The topic of the work is too broad for a single article. Due to which the individual paragraphs do not capture the topic in a holistic way and only give a cursory view of the topic. This conflicts with the phrase comprehensive in the title of the work. For example, the paper omits cadmium, nickel or zirconium nano-oxides. Hence, in my opinion, it is better to focus and describe exactly one or two nano-oxides. In addition, the authors in the paper cite review papers such as: Djurišić, A.B.; Leung, Y.H.; Ng, A.M.C.; Xu, X.Y.; Lee, P.K.H.; Degger, N.; Wu, R.S.S. Toxicity of Metal Oxide Nanoparticles: Mechanisms, Characterization, and Avoiding Experimental Artefacts. Small 2015, 11, 26-44, doi:10.1002/smll.201303947 or Liu, G.; Gao, J.; Ai, H.; Chen, X. Applications and Potential Toxicity of Magnetic Iron Oxide Nanoparticles. Small 2013, 9, 2025 1533-1545, doi:10.1002/smll.201201531. In my opinion, reference should be made to source works instead of duplicating review works. Therefore, the authors should thoroughly improve the work and resubmit it.

Reviewer 2 Report

The authors review the synthesis, preparation and biological applications of metal oxides nanoparticles. The review is well written and covers most of the relevant literature in the field. However, authors could (i) highlight the importance of the NPs in the biological area particularly, and (ii) comment on the green aspects of their synthesis. It would also be great to have a section about the future perspectives of NPs in the biological applications.

Author Response

Replies to Reviewers

We would like to thank the Reviewers for taking the time to review our manuscript and for the constructive comments which led to an improvement of the work. We have paid close attention to each point raised by the Reviewers and corrected the manuscript by considering his/her comments. Please find hereinafter our responses to the raised questions.

Reviewer 2: The authors review the synthesis, preparation and biological applications of metal oxides nanoparticles. The review is well written and covers most of the relevant literature in the field. However, authors could (i) highlight the importance of the NPs in the biological area particularly, and (ii) comment on the green aspects of their synthesis. It would also be great to have a section about the future perspectives of NPs in the biological applications.

Response: We kindly took into consideration the reviewer’s recommendations and added in the revised form of the manuscript supplementary information about the importance of metal oxide nanoparticles in the biological field. Please see the paragraph between lines 49-61. Additionally, we further emphasized the green aspects of their synthesis (lines 150-154, 249-266). Furthermore, an additional short discussion regarding future perspectives can be found in the ”Conclusions and future perspectives” section.

Reviewer 3 Report

The topic of this paper is relevant, timely, and of interest to the audience of the journal. However, there are certain points that need to be considered while revising this manuscript. I recommend the acceptance of this manuscript with major revision. Authors should consider following points while revising the manuscript:

1.     The introduction section needs revision. the authors must highlight the significance of this work by addressing the current problem statement, novelty, clear study objectives, and future direction in the introduction.

2.     Please provide an in-depth discussion by comparing the most recent literature survey on this particular area. Also, when you make a comparison with reported literature, please consider referring to the most recent works (2019 – 2023).

3.     Please address the current challenges and future direction of your study. 

4.     The importance of "green synthesis" which plays a vital role in obtaining metal nanoparticles from biological sources. Certain recent articles can be included and discussed in the manuscript as mentioned below:

https://www.tandfonline.com/doi/abs/10.1080/10667857.2020.1863571

https://www.mdpi.com/1422-0067/23/18/10626

5.     This manuscript contains some technical and grammatical mistakes; the authors must go for a thorough technical and language check 

Round 2

Reviewer 1 Report

The manuscript has been revised as recommended. In its current form, the manuscript can undergo further publishing procedures.

Reviewer 3 Report

The manuscript has been well refined as per the reviewer comments and is suitable for publication.